# Beyond the Best: Estimating Distribution Functionals in Infinite-Armed Bandits

**Yifei Wang**
Department of Electrical Engineering
Stanford University
Stanford, CA 94305
wangyf18@stanford.edu

**Tavor Z. Baharav**
Department of Electrical Engineering
Stanford University
Stanford, CA 94305
tavorb@stanford.edu

**Yanjun Han**
Institute for Data, Systems, and Society
Massachusetts Institute of Technology
Cambridge, MA 02142
yjhan@mit.edu

**Jiantao Jiao**
Department of Electrical Engineering and Computer Sciences and Department of Statistics
University of California, Berkeley
Berkeley, CA 94720
jiantao@eecs.berkeley.edu

**David Tse**
Department of Electrical Engineering
Stanford University
Stanford, CA 94305
dntse@stanford.edu

## Abstract

In the infinite-armed bandit problem, each arm's average reward is sampled from an unknown distribution, and each arm can be sampled further to obtain noisy estimates of the average reward of that arm. Prior work focuses on identifying the best arm, i.e., estimating the maximum of the average reward distribution. We consider a general class of distribution functionals beyond the maximum, and propose unified meta algorithms for both the offline and online settings, achieving optimal sample complexities. We show that online estimation, where the learner can sequentially choose whether to sample a new or existing arm, offers no advantage over the offline setting for estimating the mean functional, but significantly reduces the sample complexity for other functionals such as the median, maximum, and trimmed mean. The matching lower bounds utilize several different Wasserstein distances. For the special case of median estimation, we identify a curious thresholding phenomenon on the indistinguishability between Gaussian convolutions with respect to the noise level, which may be of independent interest.

## 1 Introduction

In the infinite-armed bandit problem (Berry et al., 1997), at each time instance the learner can either sample an arm that has been previously observed, or sample from a new arm, whose average reward

36th Conference on Neural Information Processing Systems (NeurIPS 2022).

is drawn from an unknown distribution $F$. The learner's goal is to identify arms with large average reward, with the objective being achieving either small cumulative regret (Berry et al., 1997; Wang et al., 2008; Bonald and Proutiere, 2013), or small simple regret (Carpentier and Valko, 2015). This setting differs from the classical multi-armed bandit formulation as the number of observed arms is not fixed a priori and needs to be carefully chosen by the algorithm.

We consider the problem of estimating some functional $g(F)$ of an underlying distribution $F$, as is illusrated in Figure 1. From this point of view, the classical infinite-armed bandit problem can be viewed as an *online* sampling algorithm to estimate the *maximum* of the distribution $F$. [1] Once we cast the infinite-armed bandit problem in this manner, it immediately suggests several additional questions. For example, what about *offline* sampling algorithms? Indeed, online sampling requires continual interactions with the environment which may be infeasible in certain applications, and recent work in online and offline reinforcement learning have demonstrated the significant value of both formulations (Rashidinejad et al., 2021; Zhang et al., 2021; Schrittwieser et al., 2021). Additionally, it is worth estimating functionals beyond the maximum: in many practical scenarios, including mean estimation in single-cell RNA-sequencing (Zhang et al., 2020) and Benjamini Hochberg (BH) threshold estimation in multiple hypothesis testing (Zhang et al., 2019), we are interested in the mean, median (quantile), or trimmed mean of the underlying distribution $F$. The estimation of quantiles is similar to estimation of the BH threshold, as both depend on the order statistics of the underlying distribution. Estimating the median or trimmed mean has further applications in robust statistic for instance, maintaining the fidelity of an estimator in the presence of adversarial corruption or outliers. Another natural setting where such problems arise is in large-scale distributed learning (Son and Simon, 2012). Here, a server / platform wants to estimate how much test-users like their newly released product. Users return a noisy realization of their affinity for the product, and the platform can decide to pay the user further to spend more time with the product, to test it further. For many natural objectives which are robust to a small fraction of adversarial users, e.g. trimmed mean, median, or quantile estimation, we see that our algorithm will enable estimation of the desired quantity to high accuracy while minimizing the total cost (number of samples taken). Since sampling is expensive, it is critical to identify the optimal method to collect samples, and identify the improvements afforded by adaptivity. For example, do online methods offer significant gains over offline methods? Are the fundamental limits of estimating the median and trimmed mean different from that of the maximum?

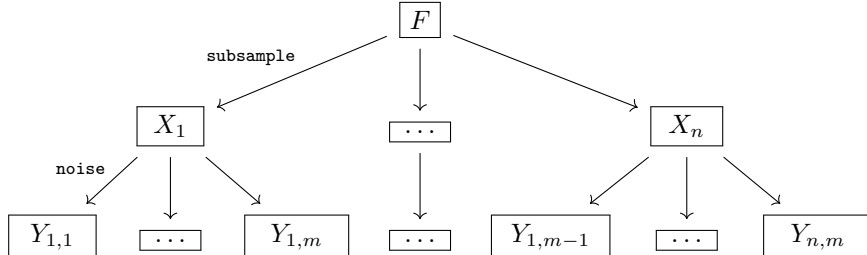

Figure 1: Problem setting. Level 0: underlying distribution $F(x)$. Level 1: unobserved samples $X_1, \ldots, X_n \sim F(x)$. Level 2: noisy observations $Y_{i,j} \sim \mathcal{N}(X_i, 1)$.

In this paper we initiate the study of distribution functional estimation in both online and offline settings and obtain both information theoretic limits and efficient algorithms for estimating the mean, median, trimmed mean, and maximum. We propose unified meta algorithms for both offline and online settings, and provide matching upper and lower boundsfor the sample complexity of estimating the aforementioned functionals in Table 1.

We also reveal new insights on the fundamental differences between the offline and online algorithms, as well as the fundamental differences between different functionals. To determine these sharp statistical limits, we use the Wasserstein-2 distance to upper bound the KL divergence in the offline setting, while the Wasserstein-$\infty$ distance is applied in the online setting instead. This approach leads to valid sample complexity lower bounds for general functionals $g$, which turn out to be tight for estimating the mean and maximum. However, a curious thresholding phenomenon, which is not

---

[1]To be precise, the objectives in infinite-armed bandit works (Berry et al., 1997; Wang et al., 2008; Bonald and Proutiere, 2013; Carpentier and Valko, 2015) are slightly different, minimizing simple or cumulative regret.

captured by the previous approach and does not occur for the *mean* and *maximum*, appears in the *median* and *trimmed mean* analyses: the KL divergence does not change smoothly with the noise level and enjoys a phase transition after the noise level exceeds some threshold. This phenomenon calls for different treatments under different estimation targets and could be of independent interest.

| Functional | Offline complexity | Online complexity | Comments |
|---|---|---|---|
| Mean | $\Theta(\varepsilon^{-2})$ | $\Theta(\varepsilon^{-2})$ | No gain from online sampling |
| Median | $\Theta(\varepsilon^{-3})$ | $\tilde{\Theta}(\varepsilon^{-2.5})$ | Holds for any quantile not on the boundary |
| Maximum | $\Theta(\varepsilon^{-(2+\beta)})$ | $\tilde{\Theta}(\varepsilon^{-\max(\beta,2)})$ | Depends on the tail regularity $\beta$ |
| Trimmed mean | $\tilde{\Theta}(\varepsilon^{-3})$ | $\tilde{\Theta}(\varepsilon^{-2.5})$ | $g(F) = \mathbb{E}\{X \mid X \in [F^{-1}(\alpha), F^{-1}(1-\alpha)]\}$ |

Table 1: Sample complexity of estimating different functionals $g(F)$, where $F$ is the cumulative distribution function (CDF) of the distribution to estimate. The trimmed mean result holds for a fixed $\alpha \in (0, 1/2)$. Here $\varepsilon$ is the target accuracy and we use $\Theta$ to denote the matching upper and lower bounds up to constants not depending on $\varepsilon$. Additionally, we use $\tilde{\Theta}, \gtrsim,$ and $\lesssim$ to suppress constants and logarithmic factors in $\varepsilon$, and $\varepsilon^c$ for any fixed $c$ arbitrarily close to zero. If $h(\varepsilon) \lesssim f(\varepsilon)$ and $f(\varepsilon) \lesssim h(\varepsilon)$ then we denote this as $f(\varepsilon) \asymp h(\varepsilon)$. For maximum estimation, we assume that the distribution satisfies $\mathbb{P}(X \geq F^{-1}(1) - \varepsilon) \asymp \varepsilon^{\beta}$. Other assumptions on $F$ are detailed in Section 3.

The rest of this paper is structured as follows. In Section 1.1 we discuss the relevant literature. We then formulate our distribution functional estimation problem in Section 2. Our unified meta algorithms for the offline and online settings are presented in Section 3, where we show the sample complexity upper bounds. We present information theoretic lower bounds proofs via Wasserstein distance for the online and offline settings in Section 4, and discuss a special thresholding phenomenon arising in median estimation in Section 5. Section 6 concludes this work.

## 1.1 Related works

The field of multi-armed bandits has seen broad interest and utility since its formalization in 1985 (Lai et al., 1985). Across clinical trials, multi-agent learning, online recommendation systems, and beyond (Lattimore and Szepesvári, 2020), multi-armed bandits have proven to be an excellent framework for modeling and solving complex tasks regarding exploration in an unknown environment. In the classical multi-armed bandit setting we have a set of $n$ distributions, where the player sequentially pulls one arm per round and observes a sample drawn from the associated reward distribution. In the infinite-armed bandit setting (Berry et al., 1997), the average arm reward for each arm is sampled i.i.d. from an unknown distribution, i.e., we have infinitely many available arms. There are many possible objectives that can be formulated in this online learning problem, from cumulative/simple regret minimization (Wang et al., 2008; Bonald and Proutiere, 2013; Carpentier and Valko, 2015; Li and Xia, 2017) to identification tasks (for example identifying an arm whose average reward is $\varepsilon$ close to the largest average reward) (Aziz et al., 2018; Chaudhuri and Kalyanakrishnan, 2017, 2019). Many works have studied best-arm identification, and we now have essentially matching instance-dependent upper and lower bounds (Jamieson and Nowak, 2014; Kaufmann et al., 2016). One could also use the average reward estimate of the identified best arm to estimate the maximum of the average reward distribution in the infinite-armed bandit setting (Carpentier and Valko, 2015; Aziz et al., 2018; Chaudhuri and Kalyanakrishnan, 2017, 2019).

From a statistical perspective, the sample complexity in the offline setting is closely related to deconvolution distribution estimation (Cordy and Thomas, 1997; Wasserman, 2004; Hall and Lahiri, 2008; Delaigle et al., 2008; Dattner et al., 2011). Nevertheless, these previous works mainly focus on the expected L2 difference between the underlying distribution function and its estimation. This simplified setting does not allow for consideration of the trade-off inherent in our setting between the number of points and the (variable) number of observations per point. Additionally, these past works did not calculate the specific sample complexity for more general functionals like quantile and trimmed mean. Since the noise is treated as fixed and uniform, there has been no study of the online setting where adaptive resampling can enable dramatic sample complexity improvements. In particular, the challenge is that we have noisy observations, which makes deriving lower bounds even in offline cases a significant challenge that has not been dealt with in the past, let alone analyzing

the online case. The dramatic performance gains afforded by adaptive resampling for functional estimation, combined with its lack of formal study, motivates the focus of this work.

## 2 Problem formulation

We are interested in estimating the distribution functional $g(F) \in \mathbb{R}$ of an underlying distribution with cumulative distribution function (CDF) $F$. We study a class of indicator-based functionals $g$ defined as follows.

**Definition 1** (Indicator-based functionals). *The functional $g$ can be represented as*

$$g(F) = \mathbb{E}\left[X | X \in S(F)\right] \tag{1}$$

*for some set $S(F)$, where $X \sim F$. The set $S(F)$ is defined as follows:*

$$S(F) = [F^{-1}(\alpha_1), F^{-1}(\alpha_2)], 0 \leq \alpha_1 \leq \alpha_2 \leq 1. \tag{2}$$

We denote $S(F)$ by $S$ throughout this work when $F$ is clear from context. This class encompasses many natural functionals of interest, which we formulate in Table 2. In Appendix **??**, we discuss extending our results to more general functionals, and show that our approach can extend to smooth reweighting functions $h(X)$ and more complex sets $S$.

| Functional | $g(F)$ | $\alpha_1$ | $\alpha_2$ | Comment |
|:---:|:---:|:---:|:---:|:---:|
| Mean | $\mathbb{E}[X]$ | 0 | 1 | |
| Quantile | $F^{-1}(\alpha)$ | $\alpha$ | $\alpha$ | $\alpha \in (0,1)$, e.g. $\alpha = 1/2$ for median |
| Maximum | $F^{-1}(1)$ | 1 | 1 | $\alpha_1 = \alpha_2 = 0$ for minimum |
| Trimmed mean | $\mathbb{E}[X | F(X) \in [\alpha_1, \alpha_2]]$ | $\alpha$ | $1 - \alpha$ | $\alpha \in (0, 1/2)$ |

Table 2: Indicator-based functionals.

As in the infinite-armed bandit setting, we only have access to noisy observations of samples drawn from the distribution with CDF $F$. We can either choose to sample from a point $X$ which we already have some noisy observations of, or draw a new point $X$ from $F$. We then observe $Y = X + Z$, where $Z \sim \mathcal{N}(0,1)$ is independent of everything observed so far.

In this paper we characterize the online and offline sample complexities of these problems, and in Section 3 propose online and offline algorithms achieving them. For $\varepsilon > 0$ and $\delta \in (0,1)$, we call an estimator $\hat{G}$ an $(\varepsilon, \delta)$-PAC approximation of $g(F)$ if $\mathbb{P}(|\hat{G} - g(F)| > \varepsilon) \leq \delta$.

## 3 Offline and online algorithms

### 3.1 Offline estimation algorithms

We study a special class of offline algorithms, which uniformly obtain observations of the points following the underlying distribution. To be precise, based on prior information regarding the distribution in question, $F$, it will choose an appropriate number of points $n$ and number of samples per point $m$ to obtain an $(\varepsilon, \delta)$-PAC approximation of $g(F)$. Specifically, the latent variables $X_1, \ldots, X_n$ are drawn from $F$, and our observations $\{Y_{i,j}\}_{j=1}^m$ are drawn i.i.d. from $\mathcal{N}(X_i, 1)$, independently for each $i$. For $i \in [n]$, denote $\hat{X}_i = m^{-1} \sum_{j=1}^m Y_{i,j}$ as the empirical mean of the observations for arm $i$. Then, we can write $\hat{X}_i = X_i + \hat{Z}_i$ where $\hat{Z}_i \sim \mathcal{N}(0, 1/m)$, independent across $i$. Define $S_n \triangleq \{i : X_{(\lfloor \alpha_1 n \rfloor)} \leq X_i \leq X_{(\lfloor \alpha_2 n \rfloor)}\}$ as the set of arms relevant for estimating the functional $g$, and define our $n$ sample estimate of $g$ as $g_n(\hat{X}_1, \ldots, \hat{X}_n) \triangleq |S_n|^{-1} \sum_{i \in S_n} \hat{X}_i$. Here $X_{(i)}$ denotes the $i$-th order statistic, that is the $i$-th smallest entry in $X_1, \ldots, X_n$. Then, $G_{n,m} = g_n(\hat{X}_1, \ldots, \hat{X}_n)$, where each $X_i$ has been sampled $m$ times, serves as a natural estimator for $g(F)$ from the noisy observations. With this, we can state the following theorem:

**Theorem 1** (Offline PAC sample complexity). *An $(\varepsilon, \delta)$-PAC offline uniform-sampling-based algorithm for estimating $g(F)$ requires $\Theta(nm)$ samples where $n, m$ depend on $\varepsilon, \delta$, the functional $g$, and information about $F$, with orderwise dependence on $\varepsilon$ detailed in Table 3.*

For the rest of this section, we discuss in greater detail our assumptions on the underlying distribution. We defer the proofs and calculations for $n$ and $m$ to **??**, as well as discussion regarding the trimmed mean to **??**.

| Functional | $m$ | $n$ |
|---|---|---|
| Mean | $\Theta(1)$ | $\Theta(\varepsilon^{-2})$ |
| Median | $\Theta(\varepsilon^{-1})$ | $\Theta(\varepsilon^{-2})$ |
| Maximum | $\Theta(\varepsilon^{-2})$ | $\Theta(\varepsilon^{-\beta})$ |
| Trimmed mean | $\Theta\left(\varepsilon^{-1}\log\left(\varepsilon^{-1}\right)\right)$ | $\Theta(\varepsilon^{-2})$ |

Table 3: Choice of $(m, n)$ for estimating different functionals to accuracy $\varepsilon$.

### 3.1.1 Mean

To guarantee that the empirical mean is a good estimator for the true mean, we impose assumptions on the tail of the distribution $F$:

**Assumption 1.** *The distribution $F$ satisfies $\mathrm{Var}_{X \sim F}[X] \leq c$.*

Assumption 1 ensures that estimation of the mean of the distribution can be accomplished with finite samples. The following proposition gives the sample complexity of the offline algorithm.

**Proposition 1.** *Suppose that Assumption 1 is satisfied. By choosing $m = 1$ and $n \geq \delta^{-1}(1 + c)\varepsilon^{-2}$, the estimator $G_{n,m}$ is an $(\varepsilon, \delta)$-PAC approximation of $g(F)$. Thus, the offline algorithm requires $O(\varepsilon^{-2})$ samples.*

### 3.1.2 Median

For median estimation we require different assumptions than the mean, as listed below.

**Assumption 2.** *There exist constants $c_1, c_2 > 0$ such that*

- $F'(x) \geq c_1$ *for $|x - \mathrm{median}(F)| \lesssim \varepsilon$.*

- $|F''(x)| \leq c_2$ *for $|x - \mathrm{median}(F)| \lesssim \sqrt{\varepsilon}$.*

The first assumption ensures that the median of $F$ is unique. The second assumption precludes the distribution from being dumbbell-shaped (very little mass near the median), in which case estimating the true median is meaningless and can be arbitrarily difficult. The following proposition gives a suitable choice of $(n, m)$ for providing an $(\varepsilon, \delta)$-PAC approximation of $g(F)$.

**Proposition 2.** *Suppose that Assumption 2 holds. Then, by choosing $m \geq 4(c_2 + 1)/(c_1\varepsilon)$ and $n \geq 28\log(1/\delta)/(c_1\varepsilon)^2$, the estimator $G_{n,m}$ is an $(\varepsilon, \delta)$-PAC approximation of $g(F)$. Thus, the offline algorithm requires $O(\varepsilon^{-3})$ samples.*

### 3.1.3 Maximum

For maximum estimation, we require an assumption on the tail of $F$ as is common in the infinite-armed bandit literature.

**Assumption 3.** *There exist constants $0 < c_1 < c_2$ and $\beta > 0$ such that*

- $1 - F(F^{-1}(1) - t) \in [c_1 t^\beta, c_2 t^\beta]$, *for all $0 \leq t \lesssim \varepsilon$.*

This assumption is also known as the $\beta$-regularity of $F$ around $F^{-1}(1)$, see (Wang et al., 2008). We present a suitable choice of $(n, m)$ in the following proposition.

**Proposition 3.** *Suppose that Assumption 3 holds. By choosing $n \geq c_1^{-1}2^\beta\varepsilon^{-\beta}\log(2/\delta)$ and $m \geq 4\varepsilon^{-2}\log(2n/\delta)$, the estimator $G_{n,m}$ is an $(\varepsilon, \delta)$-PAC approximation of $g(F)$. Therefore, the offline algorithm requires $O(\varepsilon^{-\beta-2})$ samples.*

## 3.2 Online estimation algorithm

We now present our general algorithm (Algorithm 1), an elimination-based $(\varepsilon, \delta)$-PAC algorithm that efficiently estimates $g(F)$, where $g$ is a known input functional and $F$ is an unknown distribution from which we are able to sample $X_i$ independently, and observe noisy observations $Y_{i,j}$ of $X_i$.

**Algorithm 1** `Meta Algorithm`
___
1: **Input:** target accuracy $\varepsilon$, error probability $\delta$, functional $g$ parameterized by $(\alpha_1, \alpha_2)$
2: Compute $(n, m)$ for $(\varepsilon/2, \delta/2)$-PAC estimation of $g(F)$ based on Theorem 1
3: Construct active set $A_1 = [n]$, and define $b_0 = 1$ and $t_0 = 0$
4: **for** $r = 1, 2, \ldots$ **do**
5: $\quad$ Define $b_r = 2^{-r}$ and $t_r = \min(m, \lceil 8 b_r^{-2} \log(16 n \log(m)/\delta) \rceil)$
6: $\quad$ Pull each arm in $A_r$ for $t_r - t_{r-1}$ times, construct $\hat{\mu}(r)$
7: $\quad$ Compute $A_{r+1} = \left\{ i : |\hat{\mu}_i(r) - \hat{\mu}_{(\lfloor \alpha_1 n \rfloor)}(r)| \le b_r \text{ or } |\hat{\mu}_i(r) - \hat{\mu}_{(\lfloor \alpha_2 n \rfloor)}(r)| \le b_r \right\}$
8: $\quad$ **if** $t_r \equiv m$ **then**
9: $\quad\quad$ **Break**, exit For loop
10: $\quad$ **end if**
11: **end for**
12: Construct $\hat{S}_n = \{ i : \hat{\mu}_{(\lfloor \alpha_1 n \rfloor)}(r) \le \hat{\mu}_i(r) \le \hat{\mu}_{(\lfloor \alpha_2 n \rfloor)}(r) \}$
13: **if** $\alpha_1 \equiv \alpha_2$ **then**
14: $\quad$ **return** $\frac{1}{|\hat{S}_n|} \sum_{i \in \hat{S}_n} \hat{\mu}_i(r)$
15: **else**
16: $\quad$ Draw one observation from each $i \in \hat{S}_n$, construct $\tilde{\mu}_i$
17: $\quad$ **return** $\frac{1}{|\hat{S}_n|} \sum_{i \in \hat{S}_n} \tilde{\mu}_i$
18: **end if**
___

In order to exploit the Bayesian nature of the problem, we analyze the algorithm in two parts. First, we use the fact that our arms are drawn from a common distribution to find some $n, m$ as in Theorem 1 such that the plug-in estimator $G_{n,m}$ will be an $(\varepsilon/2, \delta/2)$-PAC approximation of $g(F)$. Second, we show that our adaptive algorithm is an $(\varepsilon/2, \delta/2)$-PAC approximation of $G_{n,m}$, but is able to accomplish this using significantly fewer than $n \times m$ samples.

Notationally, we denote by $\hat{\mu}(r)$ the estimated mean vector of all arms at round $r$, and denote the $i$-th entry of this vector by $\hat{\mu}_i(r)$. We have that with high probability each arm's mean estimate stays within its width $b_r = 2^{-r}$ confidence interval for each round $r$. To analyze our algorithm, we denote $\mu_1^{\text{uni}}, \ldots, \mu_n^{\text{uni}}$ as the estimates of $X_1, \ldots, X_n$ generated by the offline algorithm after sampling each arm $m$ times. Then, we see that for the offline algorithm the arms relevant for the estimation task and the corresponding $n$ sample estimator are

$$S_n = \left\{ i : \mu_{(\lfloor \alpha_1 n \rfloor)}^{\text{uni}} \le \mu_i^{\text{uni}} \le \mu_{(\lfloor \alpha_2 n \rfloor)}^{\text{uni}} \right\}, \quad G_{n,m} = \frac{1}{|S_n|} \sum_{i \in S_n} \mu_i^{\text{uni}}. \tag{3}$$

Here $S_n$ indicates the arms that the offline algorithm believes are in $S$. We show that our online algorithm is able to efficiently estimate the set $S_n$ as $\hat{S}_n$, determining whether or not arms are in $S_n$, sampling these arms in $S_n$ sufficiently, and returning a plug-in estimator. By construction each arm is only pulled by the adaptive algorithm at most $m$ times, as we know from the analysis of the offline algorithm that for the utilized $n, m$, if each arm is pulled $m$ times then the output is an $(\varepsilon/2, \delta/2)$-PAC estimate of $g(F)$. Thus, the online algorithm's objective is essentially emulating the output of the offline algorithm, for which it only needs to sample any arm at most $m$ times.

Note that when $\alpha_1 \ne \alpha_2$, we have many samples $X_i$ that are within $S$, with $|S_n| \ge \lfloor n(\alpha_2 - \alpha_1) \rfloor$. In order to avoid issues of dependence, we discard all previous samples (as arms in $\hat{S}_n$ will have been sampled different numbers of times), and see that since we have $\Theta(n)$ arms in $\hat{S}_n$ we can construct a sufficiently accurate estimate by sampling each arm in $\hat{S}_n$ once. Algorithmically, we denote this as obtaining one fresh observation and constructing $\tilde{\mu}_i$.

To upper bound the sample complexity of our algorithm, we see that each arm only needs to be sampled to determine whether it is in $S_n$ or not. As we show in **??**, the number of samples $N(i)$ needed for point $X_i$ satisfies

$$N(i) \le \min \left( m, \frac{256 \log \left( \frac{16 n \log m}{\delta} \right)}{\left[ \text{dist}(X_i, \partial \, \text{Conv}(\{\mu_i^{\text{uni}} : i \in S_n\})) \right]^2} \right), \tag{4}$$

with probability at least $1 - \delta/4$ for all arms simultaneously, where $\partial A$ denotes the boundary of a set $A$, $\text{Conv}(A)$ denotes the convex hull of a set $A$, and $\text{dist}(X, A) = \min_{a \in A} |X - a|$. In the limit

as $\varepsilon \to 0$ we show that $\mu^{\text{uni}}_{(\lfloor \alpha_1 n \rfloor)} \to F^{-1}(\alpha_1)$ (similarly with $\mu^{\text{uni}}_{(\lfloor \alpha_2 n \rfloor)}$). This allows us to state the following theorem regarding the expected sample complexity of Algorithm 1 with respect to the distribution's relevant set of values $S$ rather than the estimated indices $S_n$.

**Theorem 2** (Meta algorithm). *For a functional $g$ satisfying Definition 1, Algorithm 1 provides an $(\varepsilon, \delta)$-PAC estimate of $g(F)$ with $M$ samples when given the requisite inputs. Here $m$ and $n$ are calculated as in Theorem 1, and the number of samples $M$ required satisfies*

$$\mathbb{E}[M] = O\left( n\mathbb{E}\left[ \min\left( m, \frac{\log(n/\delta)}{[\text{dist}(X, \partial S)]^2} \right) \right] \right). \tag{5}$$

The proof of this Theorem is deferred to **??**.

Evaluating this expression for different functionals under their corresponding assumptions yields the stated sample complexity upper bounds, as we show in **??**.

## 4 Lower bounds via Wasserstein distance

In this section we derive general lower bounds on the sample complexity of functional estimation for both offline and online algorithms, where two different Wasserstein distances play important roles. These Wasserstein-based lower bounds yield tight results for mean and maximum estimation.

### 4.1 General lower bounds based on Wasserstein distance

A classical technique for proving minimax lower bounds is Le Cam's two-point method (Le Cam et al., 2000): let $F_1$ and $F_2$ be two distributions with $|g(F_1) - g(F_2)| \geq 2\varepsilon$, and let $p_{\pi, F_1}$ and $p_{\pi, F_2}$ be the probability distributions of all observations queried by policy $\pi$ under the true population distributions $F_1$ and $F_2$, respectively. One version of Le Cam's two-point lower bound (Tsybakov, 2009, Theorem 2.2) gives

$$\inf_{\widehat{g}} \sup_{F \in \{F_1, F_2\}} \mathbb{P}_F(|\widehat{g} - g(F)| \geq \varepsilon) \geq \frac{1}{4} \exp\left( -D_{\text{KL}}(p_{\pi, F_1} \| p_{\pi, F_2}) \right).$$

Consequently, to construct a lower bound on the PAC sample complexity of estimating $g(F)$, it suffices to find the largest $\varepsilon$ such that there exist $F_1, F_2$ with $|g(F_1) - g(F_2)| \geq 2\varepsilon$ while $D_{\text{KL}}(p_{\pi, F_1} \| p_{\pi, F_2}) = O(1)$.

A key step in the above analysis is to upper bound the KL divergence $D_{\text{KL}}(p_{\pi, F_1} \| p_{\pi, F_2})$, which differs significantly between offline and online algorithms. For offline algorithms, the learner samples $n$ arms i.i.d. from $F$ with average rewards $X_1, \cdots, X_n \sim F$, and each arm is pulled $m$ times with Gaussian observations. Consequently, $p_{\pi, F} == (F * \mathcal{N}(0, 1/m))^{\otimes n}$, where $p^{\otimes n}$ denotes the $n$-fold product distribution and $*$ denotes the convolution operation. The following lemma presents an upper bound on the KL divergence for offline algorithms.

**Lemma 1.** *For any offline algorithm $\pi$ defined in Section 3.1, it holds that*

$$D_{\text{KL}}(p_{\pi, F_1} \| p_{\pi, F_2}) \leq \frac{mn}{2} \mathcal{W}_2^2(F_1, F_2),$$

*where $\mathcal{W}_2(P, Q)$ is the Wasserstein-2 distance defined as $\mathcal{W}_2^2(P, Q) = \inf_{\gamma \in \Gamma} \mathbb{E}_{(X,Y) \sim \Gamma}[(X - Y)^2]$, with $\Gamma$ being the class of all couplings between $P$ and $Q$.*

For online algorithms the distribution $p_{\pi, F}$ is no longer a product distribution as actions can depend on past observations. As a result, the KL divergence becomes larger, but still enjoys an upper bound based on another Wasserstein distance.

**Lemma 2.** *For any online algorithm $\pi$ which queries $T$ samples, it holds that*

$$D_{\text{KL}}(p_{\pi, F_1} \| p_{\pi, F_2}) \leq \frac{T}{2} \mathcal{W}_\infty^2(F_1, F_2),$$

*where $\mathcal{W}_\infty(P, Q)$ is the Wasserstein-$\infty$ distance: $\mathcal{W}_\infty(P, Q) = \inf_{\gamma \in \Gamma} \text{esssup}_{(X,Y) \sim \Gamma} |X - Y|$, with $\Gamma$ being the class of all couplings between $P$ and $Q$.*

As $\mathcal{W}_2(P,Q) \leq \mathcal{W}_\infty(P,Q)$, the upper bound of Lemma 2 is no smaller than that of Lemma 1, showing the stronger power of online algorithms. The following corollary is then immediate from Lemmas 1 and 2.

**Corollary 2.1.** *The sample complexity of $(\varepsilon, .1)$-PAC estimation of $g(F)$ is*

$$\Omega(1/\min\{\mathcal{W}_2^2(F_1, F_2) : F_1, F_2 \in \mathcal{F}, |g(F_1) - g(F_2)| \geq 2\varepsilon\})$$

*for offline algorithms, and is*

$$\Omega(1/\min\{\mathcal{W}_\infty^2(F_1, F_2) : F_1, F_2 \in \mathcal{F}, |g(F_1) - g(F_2)| \geq 2\varepsilon\})$$

*for online algorithms.*

In the remainder of this section, we show that Corollary 2.1 leads to tight lower bounds for mean and maximum estimations for both offline and online settings.

### 4.2 Lower bounds for mean estimation

Consider two distributions $F_1$ and $F_2$ which are Dirac masses supported on $1/2 - \varepsilon$ and $1/2 + \varepsilon$, respectively. Clearly $\mathcal{W}_2(F_1, F_2) = \mathcal{W}_\infty(F_1, F_2) = 2\varepsilon$, which is the best possible as $\mathcal{W}_2(F_1, F_2) \geq |\mathrm{mean}(F_1) - \mathrm{mean}(F_2)| \geq 2\varepsilon$. Corollary 2.1 gives the following lower bounds.

**Corollary 2.2.** *The $(\varepsilon, .1)$-PAC sample complexity for mean estimation is $\Omega(\varepsilon^{-2})$ for both offline and online algorithms.*

### 4.3 Lower bounds for maximum estimation

For maximum estimation, the Wasserstein distances $\mathcal{W}_2$ and $\mathcal{W}_\infty$ behave differently, as summarized in the following lemma. Let $\mathcal{F}_\beta$ be the class of densities satisfying Assumption 3.

**Lemma 3.** *For $\varepsilon \in (0, 1/2)$, it holds that*

$$\min\{\mathcal{W}_2(F_1, F_2) : F_1, F_2 \in \mathcal{F}_\beta, |\mathrm{max}(F_1) - \mathrm{max}(F_2)| \geq 2\varepsilon\} = O(\varepsilon^{\beta/2+1});$$
$$\min\{\mathcal{W}_\infty(F_1, F_2) : F_1, F_2 \in \mathcal{F}_\beta, |\mathrm{max}(F_1) - \mathrm{max}(F_2)| \geq 2\varepsilon\} = O(\varepsilon);$$
$$\min\{D_{\mathrm{KL}}(F_1\|F_2) : F_1, F_2 \in \mathcal{F}_\beta, |\mathrm{max}(F_1) - \mathrm{max}(F_2)| \geq 2\varepsilon\} = O(\varepsilon^\beta).$$

Note that we have included another term $D_{\mathrm{KL}}(F_1\|F_2)$ in Lemma 3 as it can provide a better lower bound than using $\mathcal{W}_\infty$ if $\beta \geq 2$, as $D_{\mathrm{KL}}(p_{\pi,F_1}\|p_{\pi,F_2}) \leq T \cdot D_{\mathrm{KL}}(F_1\|F_2)$ always holds due to the data-processing inequality (i.e. assuming that all arm rewards are independent). Consequently, we have the following corollary on the sample complexity of maximum estimation.

**Corollary 2.3.** *The $(\varepsilon, .1)$-PAC sample complexity for maximum estimation over $\mathcal{F}_\beta$ is $\Omega(\varepsilon^{-(\beta+2)})$ for offline algorithms, and $\Omega(\varepsilon^{-\mathrm{max}\{\beta,2\}})$ for online algorithms.*

## 5 Lower bounds via thresholding phenomenon

Although the Wasserstein distance-based approach in Section 4 provides general lower bounds for both offline and online algorithms, and these lower bounds are tight for mean and maximum estimation, sometimes this approach can be loose. For example, Lemma 3 shows that using the $\mathcal{W}_\infty$ distance might be looser than using the original KL divergence for maximum estimation. This section provides tight lower bounds for median estimation, revealing a curious thresholding phenomenon.

### 5.1 Thresholding phenomenon for offline algorithms

Let $\mathcal{F}$ denote the set of distributions satisfying Assumption 2. To use Le Cam's two-point method to prove lower bounds for offline algorithms for median estimation, the key quantity is:

$$\mathrm{KL}_\sigma(\varepsilon) \triangleq \min\{D_{\mathrm{KL}}(F_1 * \mathcal{N}(0, \sigma^2)\|F_2 * \mathcal{N}(0, \sigma^2)) : F_1, F_2 \in \mathcal{F}, |F_1^{-1}(1/2) - F_2^{-1}(1/2)| \geq 2\varepsilon\}.$$

Its inverse $\mathrm{KL}_\sigma^{-1}(\varepsilon)$ is referred to as the modulus of smoothness of the median with respect to the KL divergence under Gaussian convolution. The Wasserstein-based approach to upper bound $\mathrm{KL}_\sigma(\varepsilon)$

in Lemma 1 is the following: let $\mathcal{W}_{2,\sigma}(\varepsilon)$ be the counterpart of the above quantity with the KL divergence replaced by the Wasserstein-2 distance, Lemma 1 shows that

$$\mathrm{KL}_\sigma(\varepsilon) \leq \frac{\mathcal{W}_{2,\sigma}(\varepsilon)^2}{2\sigma^2} = \Theta\left(\frac{\varepsilon^{2.5}}{\sigma^2}\right), \tag{6}$$

an upper bound decreasing continuously with $\sigma$, where the proof of the last identity is presented in the Appendix. However, this upper bound is not tight, as shown in the following lemma.

**Lemma 4.** *For $\varepsilon \in (0, 1/4)$, $\mathrm{KL}_\sigma(\varepsilon)$ can be characterized as follows:*

$$\mathrm{KL}_\sigma(\varepsilon) \begin{cases} \in [C_1\varepsilon^2, C_2\varepsilon^2] & \text{if } \sigma \leq c\varepsilon^{1/2}, \\ \leq C(\theta, \kappa)\varepsilon^\kappa & \text{if } \sigma \geq \varepsilon^{1/2-\theta}, \end{cases}$$

*where $\theta \in (0, 1/4), \kappa \in \mathbb{N}$ are arbitrary fixed parameters, and $c, C_1, C_2, C(\theta, \kappa)$ are absolute constants with the last one depending only on $(\theta, \kappa)$.*

Lemma 4 shows a thresholding phenomenon as follows: when $\sigma$ increases from 0 to 1, the quantity $\mathrm{KL}_\sigma(\varepsilon)$ stabilizes at $\Theta(\varepsilon^2)$ whenever $\sigma \lesssim \varepsilon^{1/2}$; however, when $\sigma$ exceeds this threshold slightly (i.e. $\sigma \gtrsim \varepsilon^{1/2-\theta}$ for any constant $\theta > 0$), this quantity immediately drops to $o(\varepsilon^\kappa)$ for every possible $\kappa$. The main intuition behind this thresholding phenomenon is that, if $\sigma = O(\varepsilon^{1/2})$, the "bandwidth" of $F_1 - F_2$ exceeds that of $\mathcal{N}(0, \sigma^2)$, and the convolution is effectively using $\mathcal{N}(0, \sigma^2)$ as a Gaussian kernel (which preserves polynomials up to order 2) for smoothing $F_1 - F_2$ (which is second-order differentiable). In contrast, when $\sigma \gg \varepsilon^{1/2}$, the "bandwidth" of $F_1 - F_2$ could be smaller than $\mathcal{N}(0, \sigma^2)$, and the convolution is effectively using $F_1 - F_2$ as a kernel (which could preserve polynomials up to any desired order) for smoothing $\mathcal{N}(0, 1)$ (which is infinitely differentiable). Approximation theory tells us that the latter approximation error could be much smaller than the former, leading to the thresholding phenomenon. We remark that this phenomenon is not captured by using the $\mathcal{W}_2$ distance.

This thresholding behavior has an important consequence for median estimation. By Lemma 4 with $\sigma = 1/\sqrt{m}$, PAC learning requires that $m = \Omega(\varepsilon^{2\theta-1})$ for any offline algorithm, as otherwise the KL divergence could be made arbitrarily small. When $m$ is large enough, the first line of Lemma 4 then requires $n = \Omega(\varepsilon^{-2})$ to result in a large KL divergence for PAC learning, which comes from the idendity that

$$D_{\mathrm{KL}}(P^{\otimes n} \| Q^{\otimes n}) = n D_{\mathrm{KL}}(P \| Q).$$

Consequently, we have the following corollary for median estimation using offline algorithms.

**Corollary 2.4.** *Fix any $\theta > 0$. The $(\varepsilon, .1)$-PAC sample complexity for median estimation is $\Omega(\varepsilon^{-3+\theta})$ for any offline algorithm.*

### 5.2 Thresholding phenomenon for online algorithms

To prove the PAC lower bound for online algorithms, one first wonders if the same observation in Lemma 4 could still work. However, a close inspection of the proof reveals an issue: the optimizers $(F_1, F_2)$ in the definition of $\mathrm{KL}_\sigma(\varepsilon)$ are different under the regimes $\sigma = O(\varepsilon^{1/2})$ and $\sigma = \Omega(\varepsilon^{1/2-\theta})$. An online learning algorithm could first identify the right scenario and then choose a proper sample size to tackle the problem, and thus the above lower bound arguments break down.

To resolve this issue, we aim to choose a proper pair of distributions $(F_1, F_2)$ with $|\mathrm{median}(F_1) - \mathrm{median}(F_2)| \geq 2\varepsilon$, and investigate the behavior of $D_{\mathrm{KL}}(F_1 * \mathcal{N}(0, \sigma^2) \| F_2 * \mathcal{N}(0, \sigma^2))$ as a function of $\sigma$ with $(F_1, F_2)$ *fixed along the line*. The following lemma shows that, even for some fixed pair $(F_1, F_2)$, a similar thresholding phenomenon still holds for the KL divergence.

**Lemma 5.** *Fix any $\varepsilon, \theta \in (0, 1/4)$, and $\kappa \in \mathbb{N}$. There exist two distributions $F_1, F_2 \in \mathcal{F}$ with $|\mathrm{median}(F_1) - \mathrm{median}(F_2)| \geq 2\varepsilon$, and*

$$D_{\mathrm{KL}}(F_1 * \mathcal{N}(0, \sigma^2) \| F_2 * \mathcal{N}(0, \sigma^2)) \begin{cases} \in [C_1\varepsilon^{3/2}, C_2\varepsilon^{3/2}] & \text{if } \sigma \leq c\varepsilon^{1/2}, \\ \leq C(\theta, \kappa)\varepsilon^\kappa & \text{if } \sigma \geq \varepsilon^{1/2-\theta}, \end{cases}$$

*where $c, C_1, C_2, C(\theta, \kappa)$ are absolute constants with the last one depending only on $(\theta, \kappa)$.*

Compared with Lemma 4, Lemma 5 still shows a similar thresholding phenomenon for the KL divergence when $\sigma \gg \varepsilon^{1/2}$, but the KL divergence becomes larger for small $\sigma$ due to the additional constraint that $(F_1, F_2)$ is held fixed. Under the choice of $(F_1, F_2)$ in Lemma 5, each arm should be pulled at least $\Omega(\varepsilon^{2\theta-1})$ times, while $\Omega(\varepsilon^{\theta-3/2})$ arms need to be pulled in view of the first line. The following theorem makes the above intuition formal.

**Theorem 3.** *The $(\varepsilon, .1)$-PAC sample complexity for median estimation is $\Omega(\varepsilon^{-5/2+\theta})$ for any fixed $\theta > 0$ and any online algorithm.*

The formal proof of Theorem 3 is more complicated and requires an explicit computation of the KL divergence $D_{\mathrm{KL}}(p_{\pi,F_1} \| p_{\pi,F_2})$. We relegate the full proof to **??**. This thresholding phenomenon of the noise level also applies to the case of trimmed mean, which is discussed further and an analogous result is proved in Appendix **??**.

# 6 Conclusion

In this work we formulated and studied offline and online algorithms for estimating functionals of distributions. We developed unified algorithms for estimating the mean, median, maximum, and trimmed mean, providing sample complexity upper bounds. We additionally proved information theoretic lower bounds in these settings, which show that our algorithms are optimal up to $\varepsilon^c$ where $c$ is a fixed constant arbitrarily close to zero. We used different Wasserstein distances to construct information theoretic lower bounds for mean and maximum estimation, and showed how fundamentally different techniques are required for median and trimmed mean estimation. The lower bounds for median and trimmed mean estimation elucidate an interesting thresholding phenomenon of the noise level to distinguish two distributions after Gaussian convolution, which may be of independent interest. Interesting directions of future work include extending our analysis to non-indicator-based functionals, such as the BH threshold and analyzing the limiting behavior as $\delta \to \infty$.

## Acknowledgements

Yifei Wang and David Tse were partially supported by NSF Grants CCF-1909499. Tavor Z. Baharav was supported in part by the NSF GRFP and the Alcatel-Lucent Stanford Graduate Fellowship. Yanjun Han is supported by a Simons-Berkeley research fellowship and Norbert Wiener postdoctoral fellowship. Jiantao Jiao was partially supported by NSF Grants IIS-1901252, and CCF-1909499.

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
