## A  Extensions of the formulation

The formulation of the functional $g$ can be extended in several different ways. First, we note that we can extend the set $S(F)$ to a finite union of disjoint closed intervals, i.e., $S(F) = \cup_{i=1}^{k} S_i(F)$, where $S_i(F)$ is a closed interval for $i \in [k]$. This is because $\mathbb{E}[X|S(F)]$ can be estimated based on estimations of $\mathbb{E}[X|S_i(F)]$ via

$$\mathbb{E}[X|S(F)] = \frac{\mathbb{E}[X\mathbb{I}(X \in S(F))]}{\mathbb{P}(X \in S(F))} = \frac{\sum_{i=1}^{k} \mathbb{P}(X \in S_i(F))\mathbb{E}[X|S_i(F)]}{\sum_{i=1}^{k} \mathbb{P}(X \in S_i(F))}. \tag{7}$$

Observe that this definition naturally extends to cases where the distribution is continuous, where the density of $F$ at $X$ can be substituted for $\mathbb{P}(X \in S(F))$ for singleton sets $S(F)$. We can also consider a more general class of functionals

$$g(F) = \mathbb{E}\left[h(X)|X \in S(F)\right], \tag{8}$$

where $h$ is a differentiable function. However, when we take the limit $\varepsilon \to 0$, we see that for any fixed distribution $F$ and fixed function $h$ the reweighting induced by $h$ does not matter. Assuming that we knew whether $X_i \in S(F)$ for each $i$, we would simply want to sample $X_i \propto h'(X_i)\varepsilon^{-r}$ for some $r$. Since $h$ is differentiable, this is simply reweighting by a constant factor, which does not show up in our $\varepsilon$ dependence. Thus, we can safely only consider the weighting functional $h(x) = x$, which retains the central elimination aspect of this setting (determining whether a point is relevant or not). Loosely speaking, for any differentiable function $h$ and smooth and compactly supported $F$, we have that in the limit as $\varepsilon \to 0$ it degenerates to one of these settings.

## B  Results for trimmed mean

In this section, we present our upper and lower bound analysis for trimmed mean via both online and offline sampling algorithms.

### B.1  Upper bound for offline algorithms

For trimmed mean, the following statements are assumed to hold:

**Assumption 4.** *There exist constants $c_0, c_1, \ldots, c_5$ such that*

- $\int x^2 dF(x) \leq c_0$.
- $F'(x) \geq c_1$ *for* $|x - F^{-1}(\alpha)| \lesssim \varepsilon$ *and* $|x - F^{-1}(1-\alpha)| \lesssim \varepsilon$.
- $|F^{(2)}(x)| \leq c_2$ *for* $|x - F^{-1}(\alpha)| \lesssim \sqrt{\varepsilon}$ *and* $|x - F^{-1}(1-\alpha)| \lesssim \sqrt{\varepsilon}$.
- $F'(x) \leq c_3$ *for* $|x - F^{-1}(\alpha)| \lesssim \varepsilon$ *and* $|x - F^{-1}(1-\alpha)| \lesssim \varepsilon$.
- $\max\{|F^{-1}(\alpha)|, |F^{-1}(1-\alpha)|\} \leq c_4$, $\min\{|F^{-1}(\alpha)|, |F^{-1}(1-\alpha)|\} \geq c_5$.

The first assumption is to ensure that the mean and variance of $F$ is upper bounded, which is slightly stronger than the assumption for mean. The second assumption is to ensure that the $\alpha$ and $1 - \alpha$ quantiles of $F$ is well-defined. The third assumption ensures that the distribution has Lipschitz-continuous density around the quantiles. The forth assumption precludes the distributions which have lots of mass around the $\alpha$ and $1 - \alpha$ quantiles. The fifth assumption ensures that the $\alpha$ and $1 - \alpha$ quantiles are upper-bounded and bounded away from $0$. The following proposition gives the choice of $(n, m)$ to obtain the $(\varepsilon, \delta)$-PAC approximation of the trimmed mean.

**Proposition 4.** *Suppose that Assumption 4 holds. Then, by choosing $m \geq C_1 \varepsilon^{-1} \log \varepsilon^{-1}$ and $n \geq C_2 \varepsilon^{-2} \delta^{-1}$, the estimator $G_{n,m}$ is an $(\varepsilon, \delta)$-PAC approximation of $g(F)$. Here $C_1, C_2$ are constants which can be expressed by $c_0, \ldots, c_5$. Thus, the offline sampling algorithm takes overall $O(\varepsilon^{-3} \log(1/\varepsilon))$ samples.*

### B.2  Lower bounds for offline algorithms

Similar to the analysis for estimating median, we consider the following quantity

$$\mathrm{KL}_\sigma(\varepsilon) \triangleq \min\{D_{\mathrm{KL}}(F_1 * \mathcal{N}(0, \sigma^2) \| F_2 * \mathcal{N}(0, \sigma^2)) : F_1, F_2 \in \mathcal{F}, |g(F_1) - g(F_2)| \geq 2\varepsilon\}.$$

Analogously, we have the following bounds on the above quantity with respect to the magnitude of noise $\sigma$.

**Lemma 6.** *For $\varepsilon \in (0, 1/4)$, the following characterization of $\mathrm{KL}_\sigma(\varepsilon)$ holds as a function of $\sigma$:*

$$\mathrm{KL}_\sigma(\varepsilon) \begin{cases} \in [C_1\varepsilon^2, C_2\varepsilon^2] & \text{if } \sigma \leq c\varepsilon^{1/2}, \\ \leq C(\theta, \kappa)\varepsilon^\kappa & \text{if } \sigma \geq \varepsilon^{1/2-\theta}, \end{cases}$$

*where $\theta \in (0, 1/4), \kappa \in \mathbb{N}$ are arbitrary parameters, and $c, C_1, C_2, C(\theta, \kappa)$ are absolute constants with the last one depending only on $(\theta, \kappa)$.*

In the same manner, we have the following corollary.

**Corollary 3.1.** *Fix any $\theta > 0$. The $(\varepsilon, .1)$-PAC sample complexity for trimmed mean estimation is $\Omega(\varepsilon^{-3+\theta})$ for any offline algorithm.*

### B.3 Lower bounds for online algorithms

Analogous to the results for median, we start with the following lemma to give bounds of KL divergence between two distributions after the convolution.

**Lemma 7.** *Fix any $\varepsilon, \theta \in (0, 1/4)$, and $\kappa \in \mathbb{N}$. There exists two distributions $F_1, F_2 \in \mathcal{F}$ with $|g(F_1) - g(F_2)| \geq 2\varepsilon$, and*

$$D_{\mathrm{KL}}(F_1 * \mathcal{N}(0, \sigma^2) \| F_2 * \mathcal{N}(0, \sigma^2)) \begin{cases} \in [C_1\varepsilon^{3/2}, C_2\varepsilon^{3/2}] & \text{if } \sigma \leq c\varepsilon^{1/2}, \\ \leq C(\theta, \kappa)\varepsilon^\kappa & \text{if } \sigma \geq \varepsilon^{1/2-\theta}, \end{cases}$$

*where $c, C_1, C_2, C(\theta, \kappa)$ are absolute constants with the last one depending only on $(\theta, \kappa)$.*

We then show the lower bound for trimmed mean via online sampling algorithms.

**Theorem 4.** *Suppose that $\varepsilon > 0$. Denote $\mathcal{F}$ as the set of distributions satisfying Assumption 4. Consider an online algorithm $\pi$ with a fixed budget $t$ which outputs $\hat{G}$. Then, for any $\theta \in (0, 1/4)$, there exists at least one distribution $F \in \mathcal{F}$, such that*

$$\mathbb{P}(|\hat{G} - g(F)| > \varepsilon) \geq \frac{1}{4} \exp\left(-ct\varepsilon^{2.5-2\theta}\right), \tag{9}$$

*where $c > 0$ is a constant.*

## C Proofs of upper bounds for offline algorithms

### C.1 Mean

Here we present the proof of Proposition 1.

*Proof.* Let $X \sim F$ and $Z \sim \mathcal{N}(0, 1/m)$ are independent random variables. Then, we have

$$\mathbb{E}[X + Z] = \mathbb{E}[X] + \mathbb{E}[Z] = \mathbb{E}[X].$$

This implies that $g(F_m) = g(F)$ for any $m \geq 1$. Therefore, we can simply take $m = 1$. Then, we note that

$$\mathrm{Var}[X + Z] = \mathrm{Var}[X] + \mathrm{Var}[Z] \leq c + 1.$$

This implies that $\mathrm{Var}_{\hat{X} \sim F_1}[\hat{X}] \leq c + 1$. According to the Chebyshev inequality, we have

$$\mathbb{P}(|G_{n,m} - g(F_m)| \geq \varepsilon) \leq \frac{\mathrm{Var}_{\hat{X} \sim F_1}[\hat{X}]}{n\varepsilon^2} \leq \frac{c+1}{n\varepsilon^2}.$$

Therefore, by taking $n \geq \delta^{-1}(c+1)\varepsilon^{-2}$, we have

$$\mathbb{P}(|G_{n,m} - g(F_m)| \leq \varepsilon) \geq 1 - \delta.$$

Hence, it takes $mn = O(\varepsilon^{-2})$ samples to provide an $(\varepsilon, \delta)$-PAC approximation of $g(F)$. $\qquad \square$

## C.2 Median

Consider the following conditions

(A1) For $x \in \mathbb{R}$, there exists $c_1, t_1 > 0$ such that for all $t$ satisfying $0 \leq |t - x| \leq t_1$, $F'(t) \geq c_1$.

(A2) For $x \in \mathbb{R}$, there exists $c_2, t_2 > 0$ such that

$$|F'(x_1) - F'(x_2)|| \leq c_2.$$

for $x_1, x_2 \in [x - t_2, x + t_2]$.

We can view Assumption 2 as follows. Let $\eta = g(F) = F^{-1}(0.5)$. $F$ satisfies (A1) with $(\eta, c_1, t_1)$ and satisfies (A2) with $(\eta, c_2, t_2)$ while $t_1 \gtrsim \varepsilon$ and $t_2 \gtrsim \sqrt{\varepsilon}$. Denote $\rho(x) = F'(x)$. Let $\rho_m = \rho * \varphi_{1/m}$ as the pdf of the distribution of $\hat{X}_i$. We start with Lemma 8 to show that under suitable choice of $m$, $F_m$ also satisfies (A1).

**Lemma 8.** *Let $\eta = g(F)$. Assume that $F$ satisfies (A1) with $(\eta, c_1, t_1)$. Suppose that $m^{-1/2} \leq t_1/2$. Then, $F_m$ satisfies (A1) with $(\eta, c_1/4, t_1)$.*

*Proof.* It is sufficient to show that for $x \in [\eta - t_1, \eta + t_1]$, $\rho_m(x) > c_1/4$. As $m^{-1/2} < t_1/2$,

$$\int_0^{t_1} \varphi_{1/m}(x)dx \geq \int_0^{2m^{-1/2}} \varphi_{1/m}(x)dx = \int_0^2 \varphi_1(x)dx \geq 1/4.$$

Therefore, for $x \in [\eta, \eta + t_1]$, we have

$$\rho_m(x) = \int_z \varphi_{1/m}(z)\rho(x - z)dz \geq c_1 \int_0^{t_1} \varphi_{1/m}(z)dz \geq c_1/4.$$

Similarly, for $x \in [\eta - t_1, \eta]$, we have

$$\rho_m(x) = \int_z \varphi_{1/m}(z)\rho(x - z)dz \geq c_1 \int_{-t_1}^0 \varphi_{1/m}(z)dz = c_1 \int_0^{t_1} \varphi_{1/m}(z)dz \geq c_1/4.$$

This completes the proof. $\qquad\square$

To prove Proposition 2, we introduce the following proposition to give a point-wise bound on the difference between $F(x)$ and $F_m(x)$.

**Proposition 5.** *Suppose that $F$ satisfies (A2) with $(x, c_2, t_2)$ and $\sqrt{4 \log(2m^{-1/2})}m^{-1/2} \leq t_2$. Then, we have*

$$|F_m(x) - F(x)| \leq \frac{c_2 + 1}{2} m^{-1}.$$

*Proof.* With $k = \sqrt{4 \log(2m^{-1/2})}$, we have

$$\int_{-\infty}^{-km^{-1/2}} \varphi_{1/m}(y)dy = \int_{km^{-1/2}}^{\infty} \varphi_{1/m}(y)dy \leq e^{-k^2/2} \leq \frac{1}{4} m^{-1}.$$

For $|y| \leq t_2$, as $|F^{(2)}(x - y)| \leq c_2$, it follows that

$$|F(x - y) - F(x) - y\rho(x)| \leq \frac{c_2 y^2}{2}.$$

Note that $km^{-1/2} = \sqrt{4\log(2m^{-1/2})}m^{-1/2} \leq t_2$, we have

$$|F_m(x) - F(x)|$$

$$= \left| \int (F(x-y) - F(x))\varphi_{1/m}(y)dy \right|$$

$$\leq \int_{-\infty}^{-km^{-1/2}} \varphi_{1/m}(y)|F(x-y) - F(x)|dy + \int_{km^{-1/2}}^{\infty} \varphi_{1/m}(y)|F(x-y) - F(x)|dy$$

$$+ \left| \int_{-km^{-1/2}}^{km^{-1/2}} (F(x-y) - F(x) - y\rho(t))\varphi_{1/m}(y)dy \right| + \left| \int_{-km^{-1/2}}^{km^{-1/2}} y\rho(t)\varphi_{1/m}(y)dy \right|$$

$$\leq \frac{m^{-1}}{2} + \int_{-km^{-1/2}}^{km^{-1/2}} |F(x-y) - F(x) - y\rho(x)|\varphi_{1/m}(y)dy$$

$$\leq \frac{m^{-1}}{2} + \frac{c_2}{2} \int_{-km^{-1/2}}^{km^{-1/2}} y^2\varphi_{1/m}(y)dy \leq \frac{c_2+1}{2}m^{-1}.$$

This completes the proof. $\qquad\square$

We restate Proposition 2 as follows and present the proof.

**Proposition 6.** *Suppose that $\delta \in (0,1)$. Assume that (A1) holds at $\eta$ with $(c_1, t_1)$ and (A2) holds at $\eta$ with $(c_2, t_2)$. Suppose that $t_1 \geq \varepsilon/2$ and $t_2 \gtrsim \sqrt{\varepsilon}$. Then, with $m \geq \frac{4(c_2+1)\varepsilon^{-1}}{c_1}$ and $n \geq \frac{28\varepsilon^{-2}\log\delta^{-1}}{c_1^2}$, $G_{n,m}$ is an $(\varepsilon, \delta)$-PAC approximation of $g(F)$.*

*Proof.* Suppose that we use $n$ points and $m$ samples per point. From our choice of $m$, we have

$$\sqrt{4\log(2m^{1/2})}m^{-1/2} \leq t_2, \quad \frac{c_2+1}{2}m^{-1} \leq \frac{c_1\varepsilon}{8}.$$

Let $\eta_m = g(F_m)$ and $\eta = g(F)$. From Proposition 5, we have

$$|F_m(\eta_m) - F_m(\eta)| = |F(\eta) - F_m(\eta)| \leq c_1\varepsilon/8.$$

From Lemma 8, we note that $F_m$ satisfies (A1) with $(\eta, c_1/4, t_1)$. If $|\eta_\sigma - \eta| \geq t_1$, then, we have

$$|F_m(\eta_\sigma) - F_m(\eta)| \geq \min\{|F_m(\eta + t_1) - F_m(\eta)|, |F_m(\eta - t_1) - F_m(\eta)|\} \geq \frac{c_1 t_1}{4},$$

which leads to a contradiction. Therefore, we have

$$c_1\varepsilon/8 \geq |F_m(\eta_\sigma) - F_m(\eta)| \geq c_1|\eta_\sigma - \eta|/4.$$

This implies that $|\eta - \eta_m| \leq \varepsilon/2$. As $\varepsilon \leq t_1/2$, we note that $F_m$ satisfies (A1) with $(\eta_m, c_1/4, t_1/2)$. From the choice of $n$, according to Lemma 13, we have

$$\mathbb{P}(|G_{n,m} - \eta_m| \leq \varepsilon/2) \geq 1 - \delta.$$

Under the event $|G_{n,m} - \eta_m| \leq \varepsilon/2$, we have

$$|G_{n,m} - \eta| \leq |G_{n,m} - \eta_m| + |\eta_m - \eta| \leq \varepsilon.$$

This completes the proof. $\qquad\square$

### C.3 Maximum

In this case, the estimator for the noiseless samples writes $G_n = \max_{i \in [n]} X_n$. We first show that for sufficiently large $n$, $F(G_n)$ can be close to 1.

**Proposition 7.** *Suppose that $\varepsilon > 0$ and $\delta \in (0,1)$. Then, for $n \geq \varepsilon^{-1}\log(2/\delta)$, we have $\mathbb{P}(F(G_n) \geq 1 - \varepsilon) \geq 1 - \delta/2$.*

*Proof.* Consider a fixed number of points $n$. Note that $G_n = \max_{i \in [n]} X_i$. Therefore, we have

$$\mathbb{P}(F(G_n) \leq 1 - \varepsilon) = \mathbb{P}(F(X_i) \leq 1 - \varepsilon, \forall i \in [n])$$

$$= (1-\varepsilon)^n \leq \exp(\varepsilon^{-1}\log(2/\delta)\log(1-\varepsilon)) \leq \delta/2.$$

Here we utilize that $\log(1-\varepsilon) \leq -\varepsilon$. This completes the proof. $\qquad\square$

Then, based on the $\beta$-regularity of $F$, we show that $G_n$ can be close to $g(F)$ when $n$ is large.

**Proposition 8.** *Let $\varepsilon > 0$ and $\delta \in (0, 1)$. Denote $\eta = g(F)$. Suppose that Assumption 3 holds. Then, with $n = c_1^{-1}\varepsilon^{-\beta}\log(2/\delta)$ points, we have $\mathbb{P}(|G_n - \eta| \le \varepsilon) \ge 1 - \delta/2$.*

*Proof.* From Proposition 7, we note that

$$\mathbb{P}(F(G_n) \ge 1 - c_1\varepsilon^{\beta}) \ge 1 - \delta/2.$$

According to Assumption 3, $F(G_n) \ge 1 - c_1\varepsilon^{\beta}$ implies that $G_n \ge \eta - \varepsilon$. As $G_n = \max_{i \in [n]} X_i \le \eta$, this completes the proof. $\square$

We first choose $n \ge c_1^{-1}(\varepsilon/2)^{-\beta}\log(2/\delta)$. From Proposition 8, this guarantees that $\mathbb{P}(|G_n - \eta| \le \varepsilon/2) \ge 1 - \delta/2$. Then, by choosing $m \ge 4\varepsilon^{-2}\log(2n\delta^{-1})$, we have

$$\mathbb{P}(|X_i - \hat{X}_i| \le \varepsilon/2) \ge 1 - e^{m^{-1}\varepsilon^{-2}} \ge 1 - \delta/(2n).$$

Here we utilize the tail bound of Gaussian distributions and the fact that $X_i - \hat{X}_i \sim \mathcal{N}(0, 1/m)$. As $G_n = \max_{i \in [n]} X_i$ and $G_{n,m} = \max_{i \in [n]} \hat{X}_i$, conditioned on $\{|X_i - \hat{X}_i| \le \varepsilon/2, \forall i \in [n]\}$, we have $|G_n - G_{n,m}| \le \varepsilon/2$. Therefore, it follows that

$$\mathbb{P}(|G_n - G_{n,m}| \le \varepsilon/2) \ge P\left(|X_i - \hat{X}_i| \le \varepsilon/2, \forall i \in [n]\right) \ge 1 - n\delta/(2n) = 1 - \delta/2.$$

In summary, we have $\mathbb{P}(|\eta - G_{n,m}| \le \varepsilon) \ge \mathbb{P}(|G_n - G_{n,m}| \le \varepsilon/2) + \mathbb{P}(|G_n - \eta| \le \varepsilon/2) - 1 \ge 1 - \delta$.

### C.4 Trimmed mean

Consider the following conditions

(B1) There exists constant $c_0 > 0$ such that $\int_{-\infty}^{\infty} x^2 dF(x) \le c_0$.

(B2) For $x \in \mathbb{R}$, there exists $c_1, t_1 > 0$ such that for all $t$ satisfying $0 \le |t - x| \le t_1$, $F'(t) \ge c_1$.

(B3) For $x \in \mathbb{R}$, there exists $c_2, t_2 > 0$ such that for $x_1, x_2 \in [x - t_2, x + t_2]$,
$$|F'(x_1) - F'(x_2)| \le c_2.$$

(B4) For $x \in \mathbb{R}$, there exists $c_3, t_3 > 0$ such that for all $t$ satisfying $0 \le |t - x| \le t_3$, $F'(t) \le c_3$.

(B5) There exists constants $c_4, c_5 > 0$ such that $\max\{|F^{-1}(\alpha)|, |F^{-1}(1 - \alpha)|\} \le c_4$, $\min\{|F^{-1}(\alpha)|, |F^{-1}(1 - \alpha)|\} \ge c_5$.

We can view Assumption 4 as follows. $F$ satisfies (B1) with $c_0$ and (B5) with $c_4, c_5$. At $F^{-1}(\alpha)$ and $F^{-1}(1 - \alpha)$, $F$ satisfies (B2) with $(c_1, t_1)$, satisfies (B3) with $(c_2, t_2)$ and satisfies (B4) with $(c_3, t_3)$. Here $t_1, t_3 \gtrsim \varepsilon$ and $t_2 \gtrsim \sqrt{\varepsilon}$.

We first show that for $n \ge O(\varepsilon^{-2})$, the empirical estimator of the trimmed mean from noiseless samples will be close to the trimmed mean.

**Proposition 9.** *Suppose that Assumption 4 holds. Let $\delta \in (0, 1)$. Suppose that $\varepsilon > 0$ is sufficiently small. For*
$$n \ge (2\alpha - 1)^{-2}(4c_3c_4 + 1)^{-2}\varepsilon^{-2}\max\{16c_0\delta^{-1}, 4\log(4/\delta)c_1^{-2}\},$$
*with probability at least $1 - \delta$, we have*
$$\left|\frac{1}{n}\sum_{i=\lfloor \alpha n\rfloor}^{\lfloor(1-\alpha)n\rfloor} X_{(i)} - \int_{F^{-1}(\alpha)}^{F^{-1}(1-\alpha)} xdF(x)\right| \le \varepsilon.$$

We defer the proof to Appendix C.4.1. Then, we prove for the noisy case.

**Lemma 9.** *Suppose that Assumption 4 holds. Then, we have*
$$\left|\int_{F^{-1}(\alpha)}^{F^{-1}(1-\alpha)} xdF(x) - \int_{F^{-1}(\alpha)}^{F^{-1}(1-\alpha)} xdF_m(x)\right|$$
$$\le m^{-1}\left(4\sqrt{c_0} + k^2(2c_2c_4 + 2c_3) + 4 + 2\sqrt{c_0}c_2k^2m^{-1/2}\right) = O(m^{-1}\log(m)),$$
*where $k = \sqrt{2\log(m/2)}$.*

We leave the proof in Appendix C.4.2. From the median proof, analogously, we also have

$$|F^{-1}(\alpha) - F_m^{-1}(\alpha)| \le O(m^{-1}\log(m)) \quad , \quad |F^{-1}(1-\alpha) - F_m^{-1}(1-\alpha)| \le O(m^{-1}\log(m)).$$

Then, we have the bound

$$\left| \int_{F_m^{-1}(\alpha)}^{F_m^{-1}(1-\alpha)} x dF_m(x) - \int_{F^{-1}(\alpha)}^{F^{-1}(1-\alpha)} x dF(x) \right|$$

$$\le \left| \int_{F^{-1}(\alpha)}^{F^{-1}(1-\alpha)} x dF_m(x) - \int_{F^{-1}(\alpha)}^{F^{-1}(1-\alpha)} x dF(x) \right|$$

$$+ \left| \int_{F^{-1}(\alpha)}^{F^{-1}(1-\alpha)} x dF_m(x) - \int_{F_m^{-1}(\alpha)}^{F_m^{-1}(1-\alpha)} x dF_m(x) \right|$$

$$\le O(m^{-1}\log(m)).$$

Here we utilize that $|x|F_m'(x)$ is upper bounded. Therefore, by choosing $m = O(\varepsilon^{-1}\log(1/\varepsilon))$, we have

$$\left| \int_{F_m^{-1}(\alpha)}^{F_m^{-1}(1-\alpha)} x dF_m(x) - \int_{F^{-1}(\alpha)}^{F^{-1}(1-\alpha)} x dF(x) \right| \le \varepsilon/2.$$

By choosing $\varepsilon$ sufficiently small, $F_m$ also satisfies Assumption 4 with constants $(2c_0, c_1/2, 2c_2, 2c_3, 2c_4, c_5/2)$. Therefore, with $n \ge O(\varepsilon^{-2}\delta^{-1})$, we have

$$\mathbb{P}\left( \left| \int_{F_m^{-1}(\alpha)}^{F_m^{-1}(1-\alpha)} x dF_m(x) - G_{m,n} \right| \le \varepsilon/2 \right) \ge 1 - \delta.$$

This completes the proof.

### C.4.1  Proof of Proposition 9

*Proof.* For $\xi > 0$, denote the event

$$E(\xi) = \{|X_{(\lfloor \alpha n \rfloor)} - F^{-1}(\alpha)| \le \xi, |X_{(\lfloor (1-\alpha)n \rfloor)} - F^{-1}(1-\alpha)| \le \xi\}.$$

Choose $\xi < c_5$. From Lemma 14, with $n \ge 4\log(4/\delta)c_1^{-2}\xi^{-2}$, we have $\mathbb{P}(E(\xi)) \ge 1 - \delta/2$. Conditioned on $E(\xi)$, we note that

$$\frac{1}{n}\sum_{i=\lfloor \alpha n \rfloor}^{\lfloor (1-\alpha)n \rfloor} X_{(i)} \ge \frac{1}{n}\sum_{i \in [n]} X_i \mathbb{I}(X_i \in [F^{-1}(\alpha) + \texttt{sign}(F^{-1}(\alpha))\xi, F^{-1}(1-\alpha) - \texttt{sign}(F^{-1}(1-\alpha))\xi]),$$

and

$$\frac{1}{n}\sum_{i=\lfloor \alpha n \rfloor}^{\lfloor (1-\alpha)n \rfloor} X_{(i)} \le \frac{1}{n}\sum_{i \in [n]} X_i \mathbb{I}(X_i \in [F^{-1}(\alpha) - \texttt{sign}(F^{-1}(\alpha))\xi, F^{-1}(1-\alpha) + \texttt{sign}(F^{-1}(1-\alpha))\xi]).$$

We introduce the following lemma to show the convergence of trimmed mean.

**Lemma 10.** *Let $a < b$. Then, with $n \ge c_0 \varepsilon^{-2}\delta^{-1}$, we have*

$$\mathbb{P}\left( \left| \int_a^b x dF(x) - \frac{1}{n}\sum_{i=1}^n X_i \mathbb{I}(X_i \in [a,b]) \right| \le \varepsilon \right) \ge 1 - \delta.$$

*Proof.* Denote $Y_i = X_i \mathbb{I}(X_i \in [a,b])$. Then, $\mathbb{E}[Y_i] = \int_a^b x dF(x)$. Note that $\text{Var}[Y_i] \le \int_a^b x^2 dF(x) \le \int_{-\infty}^\infty x^2 dF(x) \le c_0$. Therefore, from the Chebyshev inequality, we have

$$P\left( \left| \int_a^b x dF(x) - \frac{1}{n}\sum_{i=1}^n X_i \mathbb{I}(X_i \in [a,b]) \right| \ge \varepsilon \right) \le \frac{c_0}{n\varepsilon^2} \le \delta.$$

This completes the proof. $\square$

From Lemma 10, for $n \geq 16c_0\xi^{-2}\delta^{-1}$, with probability at least $1 - \delta/4$, we have

$$\frac{1}{n}\sum_{i\in[n]} X_i\mathbb{I}(X_i \in [F^{-1}(\alpha) + \mathtt{sign}(F^{-1}(\alpha))\xi, F^{-1}(1-\alpha) - \mathtt{sign}(F^{-1}(1-\alpha))\xi])$$

$$\geq \int_{F^{-1}(\alpha)+\mathtt{sign}(F^{-1}(\alpha))}^{F^{-1}(1-\alpha)-\mathtt{sign}(F^{-1}(\alpha))} xdF(x) - \xi \geq \int_{F^{-1}(\alpha)}^{F^{-1}(1-\alpha)} xdF(x) - (4c_3c_4 + 1)\xi,$$

and

$$\frac{1}{n}\sum_{i\in[n]} X_i\mathbb{I}(X_i \in [F^{-1}(\alpha) - \mathtt{sign}(F^{-1}(\alpha))\xi, F^{-1}(1-\alpha) + \mathtt{sign}(F^{-1}(1-\alpha))\xi])$$

$$\leq \int_{F^{-1}(\alpha)-\mathtt{sign}(F^{-1}(\alpha))}^{F^{-1}(1-\alpha)+\mathtt{sign}(F^{-1}(\alpha))} xdF(x) + \xi \leq \int_{F^{-1}(\alpha)}^{F^{-1}(1-\alpha)} xdF(x) + (4c_3c_4 + 1)\xi.$$

Here we utilize that $\xi \leq c_5 \leq c_4$ and $|xF'(x)| \leq 2c_3c_4$ around $F^{-1}(\alpha)$ and $F^{-1}(1-\alpha)$. Combining the above bound with (C.4.1) and (C.4.1), with probability at least $1 - 3\delta/4$, we have

$$\left| \frac{1}{n}\sum_{i=\lfloor\alpha n\rfloor}^{\lfloor(1-\alpha)n\rfloor} X_{(i)} - \int_{F^{-1}(\alpha)}^{F^{-1}(1-\alpha)} xdF(x) \right| \leq (4c_3c_5 + 1)\xi.$$

Therefore, by letting $\xi = \frac{1}{(2\alpha-1)(4c_3c_5+1)}\varepsilon$, we complete the proof. $\qquad\square$

### C.4.2 Proof of Lemma 9

*Proof.* Denote $\sigma = 1/\sqrt{m}$. We also denote $F_\sigma =: F_m$. According to the Cauchy-Schwartz inequality, we have

$$\left(\int |x|dF(x)\right)^2 \leq \left(\int x^2 dF(x)\right)\left(\int 1 dF(x)\right) \leq c_0,$$

which implies that $\int |x|dF(x) \leq \sqrt{c_0}$. Let $k > 0$ be a constant. Note that

$$\int_{k\sigma}^\infty \varphi_{\sigma^2}(x)dx = \int_k^\infty \varphi_1(x)dx, \quad \int_{k\sigma}^\infty x\varphi_{\sigma^2}(x)dx = \sigma\int_k^\infty x\varphi_1(x)dx.$$

It follows that

$$\int_k^\infty x\varphi_1(x)dx = \frac{1}{\sqrt{2\pi}}\int_k^\infty e^{-\frac{x^2}{2}}d\frac{x^2}{2} = \frac{1}{\sqrt{2\pi}}e^{-k^2/2}.$$

We note that $\int_k^\infty \varphi_1(x)dx \leq e^{-k^2/2}$. By taking $k = \sqrt{-2\log(\sigma^2/2)}$, then, we have

$$\int_{k\sigma}^\infty \varphi_{\sigma^2}(x)dx \leq e^{-k^2/2} \leq \frac{1}{2}\sigma^2, \quad \int_{k\sigma}^\infty x\varphi_{\sigma^2}(x)dx \leq \sigma e^{-k^2/2} \leq \frac{1}{2}\sigma^2.$$

We can compute that

$$\left| \int_a^b xdF_\sigma(x) - \int_a^b yF'(y)dy \right|$$

$$= \left| \int_{a\leq y+z\leq b} (y+z)F'(y)\varphi_{\sigma^2}(z)dydz - \int_a^b yF'(y)dy \right|$$

$$\leq \left| \int_{a\leq y+z\leq b} yF'(y)\varphi_{\sigma^2}(z)dydz - \int_a^b yF'(y)dy \right| + \left| \int_{a\leq y+z\leq b} zF'(y)\varphi_{\sigma^2}(z)dydz \right|.$$

In the following two lemmas, we show that both terms in the last line are upper bounded by $O(\sigma^2)$.

**Lemma 11.** *We have the bound*

$$\left| \int_{a \leq y+z \leq b} yF'(y)\varphi_{\sigma^2}(z)dydz - \int_a^b yF'(y)dy \right| \leq 4\sqrt{c_0}\sigma^2 + k^2\sigma^2((|b| + |a|)c_2 + 2c_4).$$

**Lemma 12.** *We have the bound*

$$\left| \int_{a \leq y+z \leq b} zF'(y)\varphi_{\sigma^2}(z)dydz \right| \leq 4\sigma^2 + 2\sqrt{c_0}c_2k^2\sigma^3.$$

The proofs are left in Appendix C.4.3 and C.4.4. In summary, we have the bound

$$\left| \int_a^b xdF(x) - \int_a^b xdF_\sigma(x) \right|$$

$$\leq 4\sqrt{c_0}\sigma^2 + k^2\sigma^2((|b| + |a|)c_2 + 2c_4) + 4\sigma^2 + 2\sqrt{c_0}c_2k^2\sigma^3$$
$$= \sigma^2 \left( 4\sqrt{c_0} + k^2((|b| + |a|)c_2 + 2c_4) + 4 + 2\sqrt{c_0}c_2k^2\sigma \right)$$

This completes the proof. $\qquad\square$

### C.4.3   Proof of Lemma 11

We first upper-bound the LHS in (11) by the following parts:

$$\left| \int_{a \leq y+z \leq b} yF'(y)\varphi_{\sigma^2}(z)dydz - \int_a^b yF'(y)dy \right|$$

$$\leq \left| \int_{a+k\sigma}^{b-k\sigma} \left( \int_{a-y}^{b-y} \varphi_{\sigma^2}(z)dz \right) yF'(y)dzdy - \int_{a+k\sigma}^{b-k\sigma} yF'(y)dy \right| \qquad (10)$$

$$+ \left| \left( \int_{-\infty}^{a-k\sigma} + \int_{b+k\sigma}^{\infty} \right) \left( \int_{a-y}^{b-y} \varphi_{\sigma^2}(z)dz \right) yF'(y)dy \right| \qquad (11)$$

$$+ \left| \int_{a-k\sigma}^{a+k\sigma} \left( \int_{a-y}^{b-y} \varphi_{\sigma^2}(z)dz \right) yF'(y)dy - \int_a^{a+k\sigma} yF'(y)dy \right| \qquad (12)$$

$$+ \left| \int_{b-k\sigma}^{b+k\sigma} \left( \int_{a-y}^{b-y} \varphi_{\sigma^2}(z)dz \right) yF'(y)dy - \int_{b-k\sigma}^b yF'(y)dy \right|. \qquad (13)$$

For the term (10), as $y \in [a + k\sigma, b - k\sigma]$, we have $[-k\sigma, k\sigma] \subseteq [a - y, b - y]$, which implies that

$$\left| \int_{a-y}^{b-y} \varphi_{\sigma^2}(z)dz - 1 \right| \leq \left( \int_{-\infty}^{-k\sigma} + \int_{k\sigma}^{\infty} \right) \varphi_{\sigma^2}(z)dz \leq \sigma^2.$$

Hence, we have

$$\left| \int_{a+k\sigma}^{b-k\sigma} \left( \int_{a-y}^{b-y} \varphi_{\sigma^2}(z)dz \right) yF'(y)dzdy - \int_{a+k\sigma}^{b-k\sigma} yF'(y)dy \right|$$

$$\leq \sigma^2 \int_{a+k\sigma}^{b-k\sigma} |y|F'(y)dy \leq \sigma^2 \int_{-\infty}^{\infty} |y|F'(y)dy \leq \sqrt{c_0}\sigma^2.$$

For the term (11), we note that

$$\left| \left( \int_{-\infty}^{a-k\sigma} + \int_{b+k\sigma}^{\infty} \right) \left( \int_{y-a}^{y-b} \varphi_{\sigma^2}(z)dz \right) yF'(y)dy \right|$$

$$\leq \frac{1}{2}\sigma^2 \left( \int_{-\infty}^{a-k\sigma} + \int_{b+k\sigma}^{\infty} \right) |y|F'(y)dy \leq \frac{\sqrt{c_0}}{2}\sigma^2.$$

Here we utilize that for $y \geq b + k\sigma$ or $y \leq a - k\sigma$, we have

$$\int_{a-y}^{b-y} \varphi_{\sigma^2}(z)dz \leq \int_{k\sigma}^{\infty} \varphi_{\sigma^2}(z)dz \leq \frac{1}{2}\sigma^2.$$

For the term (12), we note that

$$\left| \int_{a-k\sigma}^{a+k\sigma} yF'(y)\left( \int_{a-y}^{b-y} \varphi_{\sigma^2}(z)\right) dzdy - \int_{a}^{a+k\sigma} yF'(y)dy \right|$$

$$\leq \left| \int_{0}^{k\sigma} (a+t)F'(a+t)\left( \int_{-(b-a)-t}^{b-a-t} \varphi_{\sigma^2}(z)dz\right) dt - \int_{a}^{a+k\sigma} yF'(y)dy \right|$$

$$+ \left| \int_{0}^{k\sigma} ((a-t)F'(a-t) - (a+t)F'(a+t))\left( \int_{t}^{b-a+t} \varphi_{\sigma^2}(z)\right) dzdt \right|$$

$$\leq \left| \int_{0}^{k\sigma} (a+t)F'(a+t)\left( \int_{-(b-a)-t}^{b-a-t} \varphi_{\sigma^2}(z)dz\right) dt - \int_{0}^{k\sigma} (a+t)F'(a+t)dt \right|$$

$$+ \int_{0}^{k\sigma} |a| F'(a-t) - F'(a+t)| + t|F'(a-t) + F'(a+t)|dt$$

$$\leq \sigma^2 \int_{0}^{k\sigma} |a+t|F'(a+t)dt + \int_{0}^{k\sigma} (2|a|c_2 t + 2c_4 t)\, dt$$

$$\leq \sqrt{c_0}\sigma^2 + k^2\sigma^2(|a|c_2 + c_4).$$

Similarly, for the term (13), we have the bound

$$\left| \int_{b-k\sigma}^{b+k\sigma} yF'(y)\left( \int_{a-y}^{b-y} \varphi_{\sigma^2}(z)\right) dzdy - \int_{b}^{b+k\sigma} yF'(y)dy \right| \leq \sqrt{c_0}\sigma^2 + k^2\sigma^2(|b|c_2 + c_4).$$

### C.4.4   Proof of Lemma 12

For the LHS in (12), we can decompose it into

$$\left| \int_{a \leq y+z \leq b} zF'(y)\varphi_{\sigma^2}(z)dydz \right|$$

$$\leq \left| \int_{a+k\sigma}^{b-k\sigma} F'(y)\left( \int_{a-y}^{b-y} z\varphi_{\sigma^2}(z)\right) dzdy \right| \tag{14}$$

$$+ \left| \left( \int_{-\infty}^{a-k\sigma} + \int_{b+k\sigma}^{\infty}\right) \left( \int_{a-y}^{b-y} z\varphi_{\sigma^2}(z)dz\right) F'(y)dy \right| \tag{15}$$

$$+ \left| \int_{a-k\sigma}^{a+k\sigma} F'(y)\left( \int_{a-y}^{b-y} z\varphi_{\sigma^2}(z)\right) dzdy \right| \tag{16}$$

$$+ \left| \int_{b-k\sigma}^{b+k\sigma} F'(y)\left( \int_{a-y}^{b-y} z\varphi_{\sigma^2}(z)\right) dzdy \right| \tag{17}$$

For the term (14), we note that

$$\left| \int_{a+k\sigma}^{b-k\sigma} F'(y)\left( \int_{a-y}^{b-y} z\varphi_{\sigma^2}(z)\right) dzdy \right| \leq \sigma^2 \int_{a+k\sigma}^{b-k\sigma} F'(y)dy \leq \sigma^2.$$

Here we utilize that for $y \in [a + k\sigma, b - k\sigma]$,

$$
\left| \int_{a-y}^{b-y} z\varphi_{\sigma^2}(z)dz \right| = \left| \left( \int_{-\infty}^{a-y} + \int_{b-y}^{\infty} \right) z\varphi_{\sigma^2}(z)dz \right|
$$

$$
\leq \left( \int_{-\infty}^{a-y} + \int_{b-y}^{\infty} \right) |z|\varphi_{\sigma^2}(z)dz
$$

$$
\leq \left( \int_{-\infty}^{-k\sigma} + \int_{k\sigma}^{\infty} \right) |z|\varphi_{\sigma^2}(z)dz \leq \sigma^2.
$$

We can bound the term (15) by

$$
\left| \left( \int_{-\infty}^{a-k\sigma} + \int_{b+k\sigma}^{\infty} \right) \left( \int_{a-y}^{b-y} z\varphi_{\sigma^2}(z)dz \right) F'(y)dy \right|
$$

$$
\leq \frac{1}{2}\sigma^2 \left( \int_{-\infty}^{a-k\sigma} + \int_{b+k\sigma}^{\infty} \right) F'(y)dy \leq \frac{1}{2}\sigma^2.
$$

Here we utilize that for $y \leq a - k\sigma$ or $y \geq b + k\sigma$,

$$
\left| \int_{a-y}^{b-y} z\varphi_{\sigma^2}(z)dz \right| \leq \int_{a-y}^{b-y} |z|\varphi_{\sigma^2}(z)dz \leq \int_{k\sigma}^{\infty} |z|\varphi_{\sigma^2}(z)dz \leq \frac{1}{2}\sigma^2.
$$

For the term (16), we note that

$$
\left| \int_{a-k\sigma}^{a+k\sigma} F'(y) \left( \int_{a-y}^{b-y} z\varphi_{\sigma^2}(z) \right) dzdy \right|
$$

$$
= \left| \int_0^{k\sigma} F'(a+t) \left( \int_{a-b-t}^{b-a-t} z\varphi_{\sigma^2}(z) \right) dzdt \right|
$$

$$
+ \left| \int_0^{k\sigma} (F'(a-t) - F'(a+t)) \left( \int_t^{b-a+t} z\varphi_{\sigma^2}(z) \right) dzdt \right|
$$

$$
\leq \sigma^2 \int_0^{k\sigma} F'(a+t)dt + \int_0^{k\sigma} 2c_2 t\sigma\sqrt{c_0}dt
$$

$$
\leq \sigma^2 + \sqrt{c_0}c_2 k^2\sigma^3.
$$

Here we utilize that for sufficiently small $\sigma$ such that $2k\sigma \leq b - a$,

$$
\left| \int_{a-b-t}^{b-a-t} z\varphi_{\sigma^2}(z) \right| \leq \int_{k\sigma}^{\infty} |z|\varphi_{\sigma^2}(z) + \int_{-\infty}^{-k\sigma} |z|\varphi_{\sigma^2}(z) \leq \sigma^2.
$$

Similarly, we can bound the term (17) by

$$
\left| \int_{b-k\sigma}^{b+k\sigma} F'(y) \left( \int_{a-y}^{b-y} z\varphi_{\sigma^2}(z) \right) dzdy \right| \leq \sigma^2 + \sqrt{c_0}c_2 k^2\sigma^3.
$$

This completes the proof.

## C.5   Auxiliary results

We introduce the following auxilary lemmas to extend the median results to quantile.

**Lemma 13.** *Let* $\alpha \in (0,1)$. *Suppose that* $0 < \varepsilon < \min\{\alpha, 1 - \alpha\}/5$ *and* $\delta \in (0,1)$. *For* $n \geq 4\varepsilon^{-2}\log(2/\delta)$ *points, with probability at least* $1 - \delta$, $\hat{\eta} = X_{(\lfloor \alpha n \rfloor)}$ *satisfies*

$$
|F(\hat{\eta}) - \alpha| \leq \varepsilon.
$$

*Proof.* Assume that $\varepsilon < \min\{\alpha, 1-\alpha\}/5$. Consider the random variable $Z_i = 1$ if $F(X_i) \leq \alpha - \varepsilon$ and $0$ otherwise. Let $Z = \sum_{i=1}^{n} Z_i$. By the Chernoff bound,

$$\mathbb{P}(F(\hat{\eta}) \leq \alpha - \varepsilon) \leq \mathbb{P}(Z \geq \alpha n) \leq \mathbb{P}(Z \geq (1 + \varepsilon/\alpha)\mathbb{E}[Z_1]) \leq \exp\left(-\frac{4\varepsilon^2 n}{15\alpha}\right).$$

On the other hand, consider the random variable $Z_i' = 1$ if $F(X_i) \geq \alpha + \varepsilon$ and $0$ otherwise. Let $Z' = \sum_{i=1}^{n} Z_i'$. According to the Chernoff bound,

$$\mathbb{P}(F(\hat{\eta}) \geq \alpha + \varepsilon) \leq \mathbb{P}(Z' \geq (1-\alpha)n) \leq \mathbb{P}(Z' \geq (1 + \varepsilon/(1-\alpha))\mathbb{E}[Z_i']) \leq \exp\left(-\frac{4\varepsilon^2 n}{15(1-\alpha)}\right).$$

In summary, we have

$$\mathbb{P}(|F(\hat{\eta}) - \alpha| \leq \varepsilon) \leq 1 - 2\exp\left(-\frac{4\varepsilon^2 n}{15}\right) \leq 1 - 2\exp\left(-\frac{\varepsilon^2 n}{4}\right).$$

Therefore, by taking $n = 4\varepsilon^{-2}\log(2/\delta)$, we have $\mathbb{P}(|F(\hat{\eta}) - \alpha| \leq \varepsilon) \leq \delta$. This completes the proof. $\qquad\square$

**Lemma 14.** *Assume that $F$ satisfies (B2) At $F^{-1}(\alpha)$ with $(c_1, t_1)$. Suppose that $\varepsilon \leq \min\{t_1, \min\{\alpha, 1-\alpha\}/(5c_1)\}$ and $\delta \in (0,1)$. With $n \geq 4\log(2/\delta)c_1^{-2}\varepsilon^{-2}$ points, we have $\mathbb{P}(|X_{(\lfloor \alpha n \rfloor)} - F^{-1}(\alpha)| \leq \varepsilon) \geq 1 - \delta$.*

*Proof.* Note that $c_1 \varepsilon \leq c_1 t_1 \leq \frac{1}{5}\min\{\alpha, 1-\alpha\}$. From Lemma 13, with $n \geq 4\log(2/\delta)c_1^{-2}\varepsilon^{-2}$, we have

$$\mathbb{P}(|F(X_{(\lfloor \alpha n \rfloor)}) - F(F^{-1}(\alpha))| \leq c_1\varepsilon) \geq 1 - \delta.$$

Let $\eta = F^{-1}(\alpha)$ and $\hat{\eta} = X_{(\lfloor \alpha n \rfloor)}$. If $|\hat{\eta} - \eta| > c_1 t_1$, as $F$ satisfies (B2) at $\eta$ with $(c_1, c_3, t_1)$, we have

$$|F(X_{(\lfloor \alpha n \rfloor)}) - F(F^{-1}(\alpha))| \geq \min\{|F(\eta + t_1) - F(\eta)|, |F(\eta - t_1) - F(\eta)|\} \geq c_1 t_1 > c_1\varepsilon,$$

which leads to a contradiction. If $|\hat{\eta} - \eta| \leq c_1 t_1$, then, in the same manner,

$$c_1\varepsilon|F(X_{(\lfloor \alpha n \rfloor)}) - F(F^{-1}(\alpha))| \geq c_1|\hat{\eta}|.$$

This implies that $|X_{(\lfloor \alpha n \rfloor)} - F^{-1}(\alpha)| \leq \varepsilon$. $\qquad\square$

## D  Proofs of upper bounds for online algorithms

In this Appendix we provide the proof of Theorem 2. As discussed, in order to exploit the Bayesian nature of the problem, we analyze the algorithm in two parts. First, we use the fact that our arms are drawn from a common distribution to find some $n, m$ as in Theorem 1 such that the plug-in estimator $G_{n,m}$ will be an $(\varepsilon/2, \delta/2)$-PAC approximation of $g(F)$. Second, we show that our adaptive algorithm is an $(\varepsilon/2, \delta/2)$-PAC approximation of $G_{n,m}$, but is able to accomplish this using significantly fewer samples. We begin by proving the correctness of our algorithm, afterwards analyzing its sample complexity.

### D.1  Correctness

To show correctness, we need to condition on the $n \times m$ matrix of observed samples $Y$, where $A_{i,j} = X_i + Z_{i,j}$, where $Z_{i,j}$ are i.i.d. $\mathcal{N}(0,1)$. We couple the randomness in the analysis of the offline and online algorithms, considering our random arm pulls for both to be jointly generated, and the same matrix $Y$ fed into each algorithm. Analyzing the online algorithm, we show that it recovers the result of the offline sampling algorithm within error $\varepsilon/2$ with probability at least $1 - \delta/4$.

Notationally, let $\mu_1^{\mathrm{uni}}, \ldots, \mu_n^{\mathrm{uni}}$ be the estimates of samples of offline sampling algorithm and online sampling algorithms with given $(m, n)$. Let $N(i)$ be the number of samples for point $X_i$ from the online algorithm.

Defining $g_n$ as the $n$-sample version of the functional $g$, we proceed by showing that the output of our algorithm is close to the output of the $n, m$ offline sampling algorithm, which is close to $g(F)$. Concretely, for our algorithm output $\hat{G}$, we have that

$$\mathbb{P}(|g(F) - \hat{G}| \geq \varepsilon) \leq \mathbb{P}(|g(F) - G_{n,m}| \geq \varepsilon/2) + \mathbb{P}(|G_{n,m} - \hat{G}| \geq \varepsilon/2).$$

We see from the previous arguments regarding offline sampling that for $n, m$ as selected, we have that

$$\mathbb{P}(|g(F) - G_{n,m}| \geq \varepsilon/2) \leq \delta/2.$$

Now all that remains is to show that the second term is small. We show that when $\alpha_1 = \alpha_2$, the online algorithm exactly recovers the output of the offline sampling algorithm on the event that the confidence intervals hold. When $\alpha_1 \neq \alpha_2$ (the case of the trimmed mean), we show that our estimate $\hat{G}$ is within $\varepsilon/2$ of $G_{n,m}$ with probability at least $1 - \delta/4$ on the event that the confidence intervals hold.

We begin by defining $\xi_1$ as the good event where our arms stay within their confidence intervals:

$$\xi_1 = \bigcap_{r \in \mathbb{N}, i \in [n]} \{|\hat{\mu}_i(r) - \mu_i^{\mathrm{uni}}| \leq b_r\}.$$

We give a lower bound on the probability of the good event $\xi$ in the following lemma.

**Lemma 15** (Confidence intervals). *The event $\xi_1$ defined in (D.1), where the confidence intervals of $\hat{\mu}_i^r$ about $\mu_i^{uni}$ hold, satisfies $\mathbb{P}(\xi_1) \geq 1 - \delta/4$.*

*Proof.* Recall that

$$t_r \geq 8 b_r^{-2} \log\left(\frac{16 n \log m}{\delta}\right).$$

With this choice of $t_r$, we have

$$\begin{aligned}
\mathbb{P}(\xi_1^c) &\leq \sum_{r \in \mathbb{N}, i \in [n]} \mathbb{P}(|\hat{\mu}_i(r) - \mu_i^{\mathrm{uni}}| > b_r) \\
&\leq n \sum_{r \in \mathbb{N}} \mathbb{P}(|\hat{\mu}_1(r) - \mu_1| \geq b_r/2) + \mathbb{P}(|\mu_1 - \mu_1^{\mathrm{uni}}| \geq b_r/2) \\
&\leq 4n \sum_{r \leq \lceil \log(m) \rceil} \mathbb{P}(|\hat{\mu}_1(r) - \mu_1| \geq b_r/2) \\
&\leq 4n \sum_{r \leq \lceil \log(m) \rceil} 2 \cdot \exp\left(-2 t_r (b_r/2)^2\right) \leq \delta/4.
\end{aligned}$$

This completes the proof. $\qquad \square$

On this good event $\xi_1$, we show that our online algorithm exactly recovers the partitioning of arms performed by the offline sampling algorithm. That is, for a given matrix of observations $Y$ overloading notation we can see that $g_n(Y)$, the output of the offline sampling algorithm, satisfies

$$g_n(Y) = \frac{1}{|S_n|} \sum_{i \in S_n} \mu_i^{\mathrm{uni}},$$

where $S_n$ is the set of relevant arms (i.e. those close to the boundary of $S$). Note that $S_n$ is a function of $g$. In the following lemma, we show that the online algorithm correctly identifies the arms in this set.

**Lemma 16.** *On the event $\xi_1$ we have that $\hat{S}_n$ is identical to $S_n$.*

The proof of this Lemma conditions on the good event where all confidence intervals hold, and shows that in this case the boundaries of $\hat{S}_n$ stay accurate throughout the course of the algorithm, and no arms are spuriously eliminated.

*Proof.* In this proof we focus on showing that $\hat{S}_n$ correctly partitions those elements smaller than $\mu^{\text{uni}}_{(\lfloor \alpha_1 n \rfloor)}$ from those greater than this threshold. Identical arguments hold for the analysis of $\mu^{\text{uni}}_{(\lfloor \alpha_2 n \rfloor)}$, which together imply the correctness of $\hat{S}_n$.

In round $r$, we use $(r)$ to represent the corresponding values before the sampling, for example, $\hat{\mu}(r)$. Let $t_i$ be the round that the $i$-th arm is eliminated from the active set. Suppose that the algorithm ends in $T$ rounds. We denote

$$G(r) = \{i : \hat{\mu}_i(r) > \hat{\mu}_{(\lfloor \alpha_1 n \rfloor)}(r) + b_{\min\{r, t_i\}}\},$$
$$L(r) = \{i : \hat{\mu}_i(r) < \hat{\mu}_{(\lfloor \alpha_1 n \rfloor)}(r) - b_{\min\{r, t_i\}}\},$$
$$U(r) = \{i : |\hat{\mu}_i(r) - \hat{\mu}_{(\lfloor \alpha_1 n \rfloor)}(r)| \le b_{\min\{r, t_i\}}\}.$$

From the definition of $G(r)$ and $L(r)$, it is easy to observe that $|L(T+1)| \le \lfloor \alpha_1 n \rfloor$ and $|G(T+1)| \le n - \lfloor \alpha_1 n \rfloor$. We note that $|U(T+1)| = 0$ and this implies that $|G(T+1)| = n - \lfloor \alpha_1 n \rfloor$ and $|L(T+1)| = \lfloor \alpha_1 n \rfloor$. From the definition of $L(T+1)$, it consists of $\lfloor \alpha_1 n \rfloor$ points with minimal $\hat{\mu}_i(T+1)$, i.e.,

$$\hat{\mu}_{(\lfloor \alpha_1 n \rfloor)}(T+1) = \max_{i \in L(T+1)} \hat{\mu}_i(T+1).$$

Conditioned on the good event $\xi_1$, we have

$$\hat{\mu}_i(T+1) - b_{t_i} \le \mu^{\text{uni}}_i \le \hat{\mu}_i(T+1) + b_{t_i}, i \in [n].$$

Note that $b_{T+1} = 0$. Therefore, for arbitrary $i \in L(T+1)$ and $j \in G(T+1)$, we have

$$\mu^{\text{uni}}_i \le \hat{\mu}_i(T+1) + b_{t_i} \le \hat{\mu}_{(\lfloor \alpha_1 n \rfloor)}(t) \le \hat{\mu}_j(T+1) - b_{t_j} \le \mu^{\text{uni}}_j.$$

Hence, the maximal element in $\{\mu^{\text{uni}}_i\}_{i \in L(T+1)}$ is $\mu^{\text{uni}}_{(\lfloor \alpha_1 n \rfloor)}$, which is the $\alpha_1$-th quantile of $\{\mu^{\text{uni}}_i\}_{i=1}^n$. This also implies that

$$\{i : \hat{\mu}_i(T+1) \ge \hat{\mu}_{(\lfloor \alpha_1 n \rfloor)}(T+1)\} = \{i : \mu^{\text{uni}}_i \ge \mu^{\text{uni}}_{(\lfloor \alpha_1 n \rfloor)}\}.$$

On the other hand, analogously, we note that

$$\{i : \hat{\mu}_i(T+1) \le \hat{\mu}_{(\lfloor \alpha_2 n \rfloor)}(T+1)\} = \{i : \mu^{\text{uni}}_i \le \mu^{\text{uni}}_{(\lfloor \alpha_2 n \rfloor)}\}.$$

By combining the above two equations together, we completes the proof. $\square$

We now split our analysis into cases. When $\alpha_1 = \alpha_2$, we see that all arms in $\hat{S}_n$ will be pulled exactly $m$ times, and so for $i \in \hat{S}_n$ we have that $\mu_i(r) = \mu^{\text{uni}}_i$ for the final round $r$. This implies that $\hat{G} = G_{n,m}$ for this given $Y$.

When $\alpha_1 \ne \alpha_2$, we have that some arms in the set $\hat{S}_n$ have not been pulled $m$ times; they were determined to be in $S_n$ using fewer samples, and removed from the active set as they did not require further sampling. Thus, we will not have that $\hat{G} = G_{n,m}$. Instead, we show that because there are so many points in $\hat{S}_n$, by sampling each of them only once and averaging the results, we obtain $\hat{G}$ which is within $\varepsilon/2$ of $G_{n,m}$ with probability at least $1 - \delta/4$.

**Lemma 17.** *On the event $\xi_1$, Algorithm 1 satisfies $\mathbb{P}(|G_{n,m} - \hat{G}| \ge \varepsilon/2) \le \delta/4$.*

*Proof.* If $\alpha_1 = \alpha_2$ then $G_{n,m} = \hat{G}$, and so the result holds trivially.

If $\alpha_1 \ne \alpha_2$, then on the good event $\xi_1$ where our confidence intervals hold, we have that our online algorithm correctly identifies $S_n(Y)$. Then,

$$\mathbb{P}(|G_{n,m} - \hat{G}| \ge \varepsilon/2 \mid \xi_1) \le 2\mathbb{P}\left(\left|\frac{1}{|S_n|}\sum_{i \in S_n}(\tilde{\mu}_i - \mu^{\text{uni}}_i)\right| \ge \varepsilon/2\right)$$
$$= 2\mathbb{P}\left(\left|\mathcal{N}\left(0, \frac{1 + 1/m}{|S_n|}\right)\right| \ge \varepsilon/2\right) \le \delta/4,$$

as this sum is normally distributed with variance decaying with $S_n$, and so for sufficiently large $n$ we have the desired result (as when $\alpha_1 \ne \alpha_2$, $|S_n| \ge \lfloor n(\alpha_2 - \alpha_1) \rfloor$). $\square$

Thus, we see that our algorithm's output $\hat{G}$ will be close to $g(F)$ with high probability.

## D.2 Sample complexity analysis

We now turn to bounding the sample complexity of our online algorithm. For simplicity, we overload $S_n$ in our analysis as $S_n = \text{Conv}(\{\mu_i^{\text{uni}} : i \in S_n\})$. Useful in this analysis will be the distance from $X$ to the boundary of $S_n$ (essentially the gap of $X$), which we define as

$$\text{dist}(X, \partial S_n) = \min\left( |X - \mu_{(\lfloor \alpha_1 n \rfloor)}^{\text{uni}}|, |X - \mu_{(\lfloor \alpha_2 n \rfloor)}^{\text{uni}}| \right)$$

To this end, we provide the following Lemma:

**Lemma 18.** *On the good event $\xi_1$, we have that a given arm $X_i$ will be pulled $N(i)$ times where*

$$N(i) \leq \min\left( m, \frac{256 \log\left( \frac{16 n \log m}{\delta} \right)}{[\text{dist}(X_i, \partial S_n)]^2} \right).$$

*Proof.* To begin, no arm can be pulled more than $m$ times by our adaptive algorithm, due to the structure of $t_r$. We now additionally see that by the construction of our $b_r$ confidence intervals, we have that On the good event where our confidence intervals hold, we see that an arm $X_i$ must be eliminated when $2b_r \leq \text{dist}(X_i, \partial S_n)$. Due to the iterative halving of $b_r$, this means that arm $i$ must be eliminated by round $r$ where $4b_r \geq \text{dist}(X_i, \partial S_n)$, and so $b_r^{-2} \leq 16 \left[ \text{dist}(X_i, \partial S_n) \right]^{-2}$. $\square$

This is to say, it cannot be pulled more than $m$ times, and if it is far from the boundary of $S_n$ then it can be determined whether it is in the set or not using many fewer samples, only scaling with $[\text{dist}(X, \partial S_n)]^{-2}$.

Thus, the total sample complexity of our online algorithm (for a given matrix of observed samples $Y$ with corresponding arm mean vector $\mu^{\text{uni}}$) is upper bounded by

$$\sum_{i=1}^{n} N(i) \leq \sum_{i=1}^{n} \min\left( m, \frac{256 \log\left( \frac{n \log m}{\delta} \right)}{[\text{dist}(X, \partial S_n)]^2} \right).$$

We know by the Glivenko-Cantelli theorem that in the limit $\mu_{(\lfloor \alpha_1 n \rfloor)}^{\text{uni}} \to F^{-1}(\alpha_1)$, but we require finite sample rates to give a useful bound.

**Lemma 19.** *For $n = \Theta(\varepsilon^{-2})$, with probability at least $1 - \delta/8$ in the randomness in $Y$, we have for $i \in \{1, 2\}$ that simultaneously*

$$|\mu_{(\lfloor \alpha_i n \rfloor)}^{\text{uni}} - F^{-1}(\alpha_i)| = O\left( \sqrt{\frac{\log(1/\delta)}{m}} \right).$$

*Denote this event as $\xi_2$.*

*Proof.* This lemma is simply a statement about the correctness of the offline sampling algorithm for estimating the $\alpha_1/\alpha_2$-th quantiles, which we have already proven. $\square$

Note that we are not conditioning on $\xi_2$ occurring in order for our algorithm to provide the correct output; we are simply utilizing this event to bound our algorithm's sample complexity.

On the good event in Lemma 19 and the good event where our algorithm correctly outputs an $\varepsilon$ accurate estimate and has sample complexity as in (18), we have that

$$N(i) \leq \min\left( m, \frac{256 \log\left( \frac{16 n \log m}{\delta} \right)}{\left[ \min\left( |X_i - \mu_{(\lfloor \alpha_1 n \rfloor)}^{\text{uni}}|, |X_i - \mu_{(\lfloor \alpha_2 n \rfloor)}^{\text{uni}}| \right) \right]^2} \right)$$

$$\leq \begin{cases} m & \text{when } \text{dist}(X, \partial S_n) \leq C\sqrt{\frac{\log(1/\delta)}{m}}, \\ \frac{1024 \log\left( \frac{16 n \log m}{\delta} \right)}{[\text{dist}(X_i, \partial S)]^2} & \text{when } \text{dist}(X, \partial S_n) > C\sqrt{\frac{\log(1/\delta)}{m}}, \end{cases}$$

$$\leq \min\left( m, \frac{\log\left( \frac{n \log m}{\delta} \right)}{[\text{dist}(X_i, \partial S)]^2} \right),$$

where $C$ is an absolute constant and $\text{dist}(X_i, \partial S)$ is defined analogously as $\text{dist}(X_i, \partial S) = \min\left( |X_i - F^{-1}(\alpha_1)|, |X_i - F^{-1}(\alpha_2)| \right)$.

We then have that our sample complexity $M$ is bounded as, conditioned on $\xi_1$ we have that

$$
\begin{aligned}
\mathbb{E}[M] &= O\left( \mathbb{E}\left[ \sum_{i=1}^{n} \min\left( m, \log\left( \frac{n \log m}{\delta} \right) [\text{dist}(X_i, \partial S_n)]^{-2} \right) \right] \right) \\
&\leq O\left( n\mathbb{E}\left[ \min\left( m, \log(n/\delta) [\text{dist}(X_i, \partial S_n)]^{-2} \right) | \xi_2 \right] + nm\mathbb{P}(\xi_2^c) \right) \\
&\overset{(a)}{\leq} O\left( n \log(n/\delta) \mathbb{E}\left[ \min\left( m, \text{dist}(X, \partial S)^{-2} \right) \right] \right).
\end{aligned}
$$

where we defined the good event $\xi_2$ as in Lemma 19 where our $\mu^{\text{uni}}_{(\lfloor \alpha_i n \rfloor)}$ are within $\sqrt{\log(1/\delta)/m}$ of their distributional values, i.e. $F^{-1}(\alpha_i)$. We utilize the fact that $n \geq m$ to simplify the sample complexity. (a) comes from that $\mathbb{P}(\xi_2^c) \leq \delta/8$ from Lemma 19 and that for events $E$ with probability greater than $1/2$ and positive random variables $X$, we have that $\mathbb{E}[X|E] \leq 2\mathbb{E}[X]$. This gives us the desired result.

**Theorem 5** (Restating Theorem 2). *Algorithm 1 succeeds in estimating $g(F)$ to within accuracy $\varepsilon$ with probability at least $1 - \delta$, and requires at most*

$$
O\left( n \log(n/\delta) \mathbb{E}\left[ \min\left( m, \text{dist}(X, \partial S)^{-2} \right) \right] \right) \tag{18}
$$

*observations in expectation.*

### D.3 Functional-specific upper bounds

From Theorem 2, we are able to derive the upper bound sampling complexity of online algorithms in Table 1 for mean, median, maximum and trimmed mean estimation by analyzing (18) under the functional specific assumptions.

#### D.3.1 Mean estimation

*Proof.* For mean estimation, from Theorem 1 we have that $n = \Theta(\varepsilon^{-2})$ and $m = \Theta(1)$ is sufficient. Therefore, we have an expected sample complexity of

$$
\begin{aligned}
\mathbb{E}[M] &= O\left( n \log(n/\delta) \mathbb{E}\left[ \min\left( m, \text{dist}(X, \partial S)^{-2} \right) \right] \right) \\
&= O(n \log(n/\delta)) = O(\varepsilon^{-2} \log(1/\varepsilon)).
\end{aligned} \tag{19}
$$

This completes the proof. $\qquad \square$

#### D.3.2 Median estimation

*Proof.* For median estimation, from Theorem 1 we have that $n = \Theta(\varepsilon^{-2})$ and $m = \Theta(\varepsilon^{-1})$ is sufficient. We can compute that

$$
\begin{aligned}
&\mathbb{E}\left[ \min\left( m, \text{dist}(X, \partial S)^{-2} \right) \right] \\
&= O(1) \int \min\left( \varepsilon^{-1}, (x - F^{-1}(0.5))^{-2} \right) dF(x) \\
&= O(1) \left( \int_{-\infty}^{F^{-1}(0.5)-\sqrt{\varepsilon}} (x - F^{-1}(0.5))^{-2} dF(x) + \int_{F^{-1}(0.5)+\sqrt{\varepsilon}}^{\infty} (x - F^{-1}(0.5))^{-2} dF(x) \right) \\
&\quad + O(1) \left( \int_{F^{-1}(0.5)-\sqrt{\varepsilon}}^{F^{-1}(0.5)+\sqrt{\varepsilon}} \varepsilon^{-1} dx \right).
\end{aligned}
$$

The first term can be bounded using integration by parts, where we note that

$$\int_{-\infty}^{F^{-1}(0.5)-\sqrt{\varepsilon}} (x - F^{-1}(0.5))^{-2} dF(x)$$

$$= -\int_{-\infty}^{F^{-1}(0.5)-\sqrt{\varepsilon}} F'(x) d(x - F^{-1}(0.5))^{-1}$$

$$= -F'(x)(x - F^{-1}(0.5))^{-1}\Big|_{-\infty}^{F^{-1}(0.5)-\sqrt{\varepsilon}} + \int_{-\infty}^{F^{-1}(0.5)-\sqrt{\varepsilon}} F^{(2)}(x)(x - F^{-1}(0.5))^{-1} dx$$

$$\leq \sqrt{\varepsilon^{-1}} F'(F^{-1}(0.5) - \sqrt{\varepsilon}) + \sqrt{\varepsilon^{-1}} \int_{-\infty}^{F^{-1}(0.5)-\sqrt{\varepsilon}} F^{(2)}(x)$$

$$\leq 2\sqrt{\varepsilon^{-1}} F'(F^{-1}(0.5) - \sqrt{\varepsilon}) = O(\sqrt{\varepsilon^{-1}}).$$

Here we utilize that $F'(x)$ is upper bounded at $F^{-1}(0.5) - \sqrt{\varepsilon}$. Similarly, we have

$$\int_{F^{-1}(0.5)+\sqrt{\varepsilon}}^{\infty} (x - F^{-1}(0.5))^{-2} dF(x) \leq O(\sqrt{\varepsilon}).$$

In summary, we have

$$\mathbb{E}\left[\min\left(m, \text{dist}(X, \partial S)^{-2}\right)\right] \leq O(\sqrt{\varepsilon^{-1}}),$$

and this implies that

$$\mathbb{E}[M] = O\left(n\log(n/\delta)\mathbb{E}\left[\min\left(m, \text{dist}(X, \partial S)^{-2}\right)\right]\right) \leq O(n\log(n/\delta)\varepsilon^{-0.5})$$
$$= O(\varepsilon^{-2.5}\log(1/\varepsilon)).$$

This completes the proof. $\qquad\square$

### D.3.3 Maximum estimation

*Proof.* For maximum estimation, from Theorem 1 we have that $n = \Theta(\varepsilon^{-\beta})$ and $m = \Theta(\varepsilon^{-2})$. We note that

$$\mathbb{E}\left[\min\left(m, \text{dist}(X, \partial S)^{-2}\right)\right]$$

$$= O(1)\int \min\left(\varepsilon^{-2}, (x - F^{-1}(1))^{-2}\right) dF(x)$$

$$= O(1)\left(\int_{-\infty}^{F^{-1}(1)-\varepsilon} (x - F^{-1}(1))^{-2} dF(x) + \int_{F^{-1}(1)-\varepsilon}^{\infty} \varepsilon^{-2} dF(x)\right).$$

For $\beta < 2$, we can compute that

$$\int_{-\infty}^{F^{-1}(1)-\varepsilon} (x - F^{-1}(1))^{-2} dF(x) = \int_{-\infty}^{F^{-1}(1)-\varepsilon} F'(x)(x - F^{-1}(1))^{-2} dx$$

$$= \int_{\varepsilon}^{\infty} F'(F^{-1}(1) - x)x^{-2} dx$$

$$\leq \int_{\varepsilon}^{\infty} c_2\beta x^{\beta-1} x^{-2} dx = \frac{c_2\beta}{2-\beta}\varepsilon^{\beta-2},$$

and

$$\int_{F^{-1}(1)-\varepsilon}^{\infty} \varepsilon^{-2} dF(x) = \varepsilon^{-2}(1 - F(F^{-1}(1) - \varepsilon)) \leq c_2\varepsilon^{\beta-2}.$$

This implies that

$$\mathbb{E}\left[\min\left(m, \text{dist}(X, \partial S)^{-2}\right)\right] = O(\varepsilon^{\beta-2}).$$

As a result, we have

$$\mathbb{E}[M] = O\left(n\log(n/\delta)\mathbb{E}\left[\min\left(m, \text{dist}(X, \partial S)^{-2}\right)\right]\right) = O(\varepsilon^{-2}\log(1/\varepsilon)).$$

For $\beta = 2$, we can compute that

$$\int_{-\infty}^{F^{-1}(1)-\varepsilon} (x - F^{-1}(1))^{-2} dF(x) = \int_{-\infty}^{F^{-1}(1)-\varepsilon} F'(x)(x - F^{-1}(1))^{-2} dx$$

$$= \int_{\varepsilon}^{F^{-1}(1)-F^{-1}(0)} F'(F^{-1}(1) - x)x^{-2} dx$$

$$\leq \int_{\varepsilon}^{F^{-1}(1)-F^{-1}(0)} 2c_2 x x^{-2} dx = O(\log \varepsilon^{-1}).$$

Hence, we have

$$\mathbb{E}[M] = O\left(n \log(n/\delta)\mathbb{E}\left[\min\left(m, \text{dist}(X, \partial S)^{-2}\right)\right]\right) \leq O(\varepsilon^{-2}\log(1/\varepsilon)).$$

For $\beta > 2$, we note that

$$\int_{-\infty}^{F^{-1}(1)-\varepsilon} (x - F^{-1}(1))^{-2} dF(x) = \int_{-\infty}^{F^{-1}(1)-\varepsilon} F'(x)(x - F^{-1}(1))^{-2} dx$$

$$= \int_{\varepsilon}^{\infty} F'(F^{-1}(1) - x)x^{-2} dx$$

$$\leq \int_{\varepsilon}^{F^{-1}(1)-F^{-1}(0)} c_2 \beta x^{\beta-1} x^{-2} dx = O(1).$$

Hence, we have

$$\mathbb{E}[M] = O\left(n \log(n/\delta)\mathbb{E}\left[\min\left(m, \text{dist}(X, \partial S)^{-2}\right)\right]\right) \leq O(\varepsilon^{-\beta}\log(1/\varepsilon)).$$

In summary, we have

$$\mathbb{E}[M] = O\left(n \log(n/\delta)\mathbb{E}\left[\min\left(m, \text{dist}(X, \partial S)^{-2}\right)\right]\right) \leq O(\varepsilon^{-\max\{\beta,2\}}\log(1/\varepsilon)).$$

This completes the proof. $\qquad\square$

### D.3.4 Trimmed mean estimation

*Proof.* For trimmed mean, we note that the analysis is similar to the case of median. This gives that

$$O\left(n \log(n/\delta)\mathbb{E}\left[\min\left(m, \text{dist}(X, \partial S)^{-2}\right)\right]\right) \leq O(n \log(n)\varepsilon^{-0.5}\log(1/\varepsilon)) = O(\varepsilon^{-2.5}\log^2(1/\varepsilon)).$$

$\qquad\square$

## E   Proofs in Section 4

### E.1   Proof of Lemma 1

*Proof.* Denote $p_{F_1}$ and $p_{F_2}$ as the pdf of $F_1$ and $F_2$ respectively. Let $\sigma = 1/\sqrt{m}$ and let $\varphi_{\sigma^2}$ be the pdf of $\mathcal{N}(0, \sigma^2)$. As $p_{\pi,F_1} = (p_{F_1} * \varphi_{\sigma^2})^{\otimes n}$ and $p_{\pi,F_2} = (p_{F_2} * \varphi_{\sigma^2})^{\otimes n}$, we have

$$D_{\text{KL}}(p_{\pi,F_1} \| p_{\pi,F_2}) = n D_{\text{KL}}(p_{F_1} * \varphi_{\sigma^2} \| p_{F_2} * \varphi_{\sigma^2}).$$

On the other hand, we note that

$$p_{F_1} * \varphi_{\sigma^2}(y) = \int_{-\infty}^{\infty} p_{F_1}(x)\varphi_{\sigma^2}(y - x)dx = \mathbb{E}_{X \sim F_1}[\varphi_{\sigma^2}(y - X)].$$

Similarly, we have $p_{F_1} * \varphi_{\sigma^2}(y) = \mathbb{E}_{X' \sim F_2}[\varphi_{\sigma^2}(y - X')]$.

Let $\gamma \in \Gamma$ be a coupling of $F_1$ and $F_2$. Namely, it is a joint distribution of $(X, X')$ and its marginal distribution on $X$ ($X'$) are $F_1$ ($F_2$). Then, utilizing the convexity of KL divergence, we have

$$D_{\text{KL}}(p_{F_1} * \varphi_{\sigma^2} \| p_{F_2} * \varphi_{\sigma^2}) = D_{\text{KL}}(\mathbb{E}_{(X,X')\sim\gamma}[\varphi_{\sigma^2}(y - X)] \| \mathbb{E}_{(X,X')\sim\gamma}[\varphi_{\sigma^2}(y - X)])$$

$$\leq \mathbb{E}_{(X,X')\sim\gamma} D_{\text{KL}}(\varphi_{\sigma^2}(y - X) \| \varphi_{\sigma^2}(y - X')) = \mathbb{E}_{(X,X')\sim\gamma} \frac{\|X - X'\|_2^2}{2\sigma^2}.$$

By taking the infimum w.r.t. all possible coupling $\gamma$, we note that

$$D_{\text{KL}}(p_{\pi,F_1} \| p_{\pi,F_2}) = n D_{\text{KL}}(F_1 * \mathcal{N}(0, \sigma^2) \| F_2 * \mathcal{N}(0, \sigma^2))$$

$$\leq n \inf_{\gamma \in \Gamma} \mathbb{E}_{(X,X')\sim\gamma} \frac{\|X - X'\|_2^2}{2\sigma^2} = \frac{mn}{2}\mathcal{W}(F_1, F_2)^2.$$

This completes the proof. $\qquad\square$

## E.2 Proof of Lemma 2

*Proof.* For a given underlying distribution $F_1$ and a given algorithm $\pi$, the joint distribution of $\{(X_i, A_i, Y_i)\}_{i=1}^t$ has the following probability density function

$$p_{\pi, F_1}(\{(x_i, a_i, y_i)\}_{i=1}^t) = \prod_{i=1}^t p_{F_1}(x_i) p_\pi(a_i | (a_j, y_j)_{j=1}^{i-1}) p(y_i | a_i, \{x_j\}_{j=1}^i).$$

Thus, we can also write

$$p_{\pi, F_1}(\{(x_i, a_i, y_i)\}_{i=1}^t) = p_\pi(\{(a_i, y_i)\}_{i=1}^t | \{x_i\}_{i=1}^t) \prod_{i=1}^t p_{F_1}(x_i),$$

where

$$p_\pi(\{(a_i, y_i)\}_{i=1}^t | \{x_i\}_{i=1}^t) = \prod_{i=1}^t p_\pi(a_i | (a_j, y_j)_{j=1}^{i-1}) p(y_i | x_{a_i}).$$

Thus, the marginal distribution on $\{(a_i, y_i)\}_{i=1}^t$ follows

$$p_{\pi, F_1}(\{(a_i, y_i)\}_{i=1}^t) = \int p_{\pi, F_1}(\{(x_i, a_i, y_i)\}_{i=1}^t) dz_1 \dots dz_t$$

$$= \mathbb{E}_{(X_i)_{i=1}^t \sim F_1}[p_\pi(\{(a_i, y_i)\}_{i=1}^t | \{X_i\}_{i=1}^t)].$$

Let $F_2$ be a distribution different from $F_1$. We want to bound the KL divergence from $p_{\pi, F_1}(\{(a_i, y_i)\}_{i=1}^t)$ to $p_{\pi, F_2}(\{(a_i, y_i)\}_{i=1}^t)$. Let $\gamma \in \Gamma$ be a joint distribution with marginals $F_1$ and $F_2$. For simplicity, we write $\mathbb{E}_\gamma = \mathbb{E}_{(X_i, X_i')_{i=1}^t \sim \gamma}$. Utilizing the convexity of KL divergence, we note that

$$D_{\mathrm{KL}}(p_{\pi, F_1} \| p_{\pi, F_2}) = D_{\mathrm{KL}}(\mathbb{E}_\gamma[p_\pi(\cdot | \{X_i\}_{i=1}^t)] \| \mathbb{E}_\gamma[p_\pi(\cdot | \{X_i'\}_{i=1}^t)])$$

$$\leq \mathbb{E}_\gamma D_{\mathrm{KL}}(p_\pi(\cdot | \{X_i\}_{i=1}^t) \| p_\pi(\cdot | \{X_i'\}_{i=1}^t)). \tag{20}$$

Given the pair of underlying states $(\{X_i\}_{i=1}^t, \{X_i'\}_{i=1}^t)$, we can compute that

$$D_{\mathrm{KL}}(p_\pi(\cdot | \{X_i\}_{i=1}^t)) \| p_\pi(\cdot | \{X_i'\}_{i=1}^t)))$$

$$= \mathbb{E}_{\{(A_i, Y_i)\}_{i=1}^t \sim p_\pi(\cdot | \{X_i\}_{i=1}^t)} \left[ \sum_{i=1}^t \log \frac{p(Y_i | X_{A_i})}{p(Y_i | X_{A_i}')} \right]$$

$$= \mathbb{E}_{\{A_i\}_{i=1}^t \sim p_\pi(\cdot | \{X_i\}_{i=1}^t)} \left[ \sum_{i=1}^t \sum_{j=1}^t \mathbb{I}(A_j = i) \frac{1}{2} |X_i - X_i'|^2 \right]$$

$$= \sum_{i=1}^t \frac{C_i(\{X_j\}_{j=1}^t)}{2} |X_i - X_i'|^2.$$

Here we let $C_i(\{X_j\}_{j=1}^t) = \mathbb{E}_{\{A_i\}_{i=1}^t \sim p_\pi(\cdot | \{X_i\}_{i=1}^t)} \left[ \sum_{j=1}^t \mathbb{I}(A_j = i) \right]$. This implies that

$$\mathbb{E}_\gamma D_{\mathrm{KL}}(p_{\pi, F_1}(\cdot | \{X_j\}_{j=1}^t) \| p_{\pi, F_2}(\cdot | \{X_i'\}_{i=1}^t)) = \mathbb{E}_\gamma \left[ \sum_{i=1}^t \frac{C_i(\{X_j\}_{j=1}^t)}{2} |X_i - X_i'|^2 \right].$$

We note that $\sum_{i=1}^t C_i(\{X_j\}_{j=1}^t) = t$ and

$$\mathbb{E}_\gamma |X_i - X_i'|^2 \leq \mathrm{esssup}_{(X, X') \sim \gamma} |X - X'|^2, \quad i = 1, \dots, t$$

This implies that

$$\mathbb{E}_\gamma D_{\mathrm{KL}}(p_{\pi, F_1}(\cdot | \{X_j\}_{j=1}^t) \| p_{\pi, F_2}(\cdot | \{X_i'\}_{i=1}^t)) \leq \frac{t}{2} \cdot \mathrm{esssup}_{(X, X') \sim \gamma} |X - X'|^2.$$

By taking the infimum w.r.t. $\gamma$, we have

$$D_{\mathrm{KL}}(p_{\pi, F_1} \| p_{\pi, F_2}) \leq \inf_{\gamma \in \Gamma} \mathbb{E}_\gamma D_{\mathrm{KL}}(p_{\pi, F_1}(\cdot | \{X_j\}_{j=1}^t) \| p_{\pi, F_2}(\cdot | \{X_i'\}_{i=1}^t))$$

$$\leq t \inf_{\gamma \in \Gamma} \mathrm{esssup}_{(X, X') \sim \gamma} \frac{|X - X'|^2}{2} = \frac{t}{2} \mathcal{W}_\infty(F_1, F_2)^2.$$

This completes the proof. $\qquad\qquad\square$

### E.3 Proof of Lemma 3

*Proof.* We first give the example for the Wasserstein-2 distance. Let $G(s)$ be defined as

$$G^{-1}(s) = \begin{cases} g_1(s), & t \in [0, 2\varepsilon], \\ s, & s \in [2\varepsilon, 1], \end{cases}$$

where $g(s)$ is a monotonic cubic interpolation satisfying that

$$g_1(0) = \varepsilon, g_1'(0) = 1, g_1(2\varepsilon) = 2\varepsilon, g_1'(2\varepsilon) = 1,$$

We note that the image of $G^{-1}$ is $[\varepsilon, 1]$. Therefore, the domain of $G$ is $[\varepsilon, 1]$. We also note that for $s \in [0, \varepsilon]$, we have

$$|G(s) - s| \leq \varepsilon.$$

Consider the following two distributions. We consider a distribution with

$$F_1(x) = 1 - (1 - x)^\beta$$

in its support $[0, 1]$ and another distribution with CDF

$$F_2(x) = 1 - (G(1 - x))^\beta$$

in its support $[0, 1 - \varepsilon]$. We can verify that $F_1$ and $F_2$ satisfy Assumption 3 and $|\max(F_1) - \max(F_2)| = \varepsilon$. We note that

$$F_1^{-1}(s) = 1 - (1 - s)^{1/\beta}, F_2^{-1}(s) = 1 - G^{-1}((1 - s)^{1/\beta}),$$

We can verify that $F_1$ and $F_2$ satisfy Assumption 3 and $|\max(F_1) - \max(F_2)| \models \varepsilon$. The Wasserstein-2 distance between $F_1$ and $F_2$ can be computed as

$$\begin{aligned}
\mathcal{W}_2(F_1, F_2)^2 &= \int_0^1 \left(F_1^{-1}(s) - F_2^{-1}(s)\right)^2 ds \\
&= \int_0^{\varepsilon^\beta} \left(s^{1/\beta} - G(s^{1/\beta})\right)^2 ds \\
&\leq \int_0^{\varepsilon^\beta} \varepsilon^2 ds = \varepsilon^{\beta+2}.
\end{aligned}$$

We then give the example for the Wasserstein-$\infty$ distance. Consider a distribution with CDF

$$F_1(x) = 1 - (1 - x)^\beta$$

in its support $[0, 1]$ and another distribution with CDF

$$F_2(x) = 1 - (1 - x + \varepsilon)^\beta$$

in its support $[\varepsilon, 1 + \varepsilon]$. Let $\gamma$ be the joint distribution of $(X, X + \varepsilon)$, where $X$ follows $F_1$. Then, $\gamma \in \Gamma$ is the coupling of $F_1$ and $F_2$. We can compute that

$$\mathrm{esssup}_{(X,X')\sim\gamma}|X - X'| = \varepsilon.$$

This implies that $\mathcal{W}_\infty(F_1, F_2) \leq \varepsilon$.

Finally, we give the example for the KL divergence. Consider two distributions with following CDFs:

$$F_1(x) = \begin{cases} \dfrac{1 - (1 - x)^\beta}{1 - \varepsilon^\beta}, & x \in [0, 1 - \varepsilon], \\ 0, & x \in (-\infty, 0), \\ 1, & x \in (1 - \varepsilon, \infty). \end{cases}$$

$$F_2(x) = \begin{cases} 1 - (1 - x)^\beta, & x \in [0, 1], \\ 0, & x \in (-\infty, 0), \\ 1, & x \in (1 - \varepsilon, \infty). \end{cases}$$

We can verify that $F_1$ and $F_2$ satisfy Assumption 3 and $|\max(F_1) - \max(F_2)| = \varepsilon$. We note that

$$\sup_{x \in [0, 1-\varepsilon]} \frac{p_{F_1}(x)}{p_{F_2}(x)} = \frac{1}{1 - \varepsilon^\beta} =: \zeta.$$

Thus, according to the reverse Pinsker inequality, we have

$$D_{KL}(F_1 \| F_2) \leq \frac{\log \zeta}{1 - \zeta^{-1}} D_{TV}(F_1 \| F_2).$$

We note that $\lim_{\varepsilon \to 0} \frac{\log \zeta}{1 - \zeta^{-1}} = \lim_{\varepsilon \to 0} \frac{-\log(1 - \varepsilon^\beta)}{\varepsilon^\beta} = 1$. For $x \in [0, 1 - \varepsilon]$.

$$F_1(x) - F_2(x) = \frac{\varepsilon^\beta}{1 - \varepsilon^\beta} F_2(x).$$

Therefore, we have

$$D_{TV}(F_1, F_2) = \max_{x \in [0,1]} (F_1(x) - F_2(x)) \leq \frac{\varepsilon^\beta}{1 - \varepsilon^\beta} = O(\varepsilon^\beta).$$

This implies that

$$D_{KL}(F_1 \| F_2) \leq O(\varepsilon^\beta).$$

This completes the proof. $\qquad \square$

## F  Proof of lower bounds for median estimation

We start with an auxiliary lemma to pointwisely bound the log-likelihood difference of two distributions.

**Lemma 20.** *Consider two densities supported on $[-1, 1]$ with pdf $p(x)$ and $q(x)$ such that $p(x), q(x) \geq 1/4$ and $|p(x) - q(x)| \leq \varepsilon \cdot \mathbb{1}(|x| \leq \zeta)$ for all $x \in [-1, 1]$, and $\int_{-\zeta}^{\zeta} x^\ell (p(x) - q(x)) dx = 0$ for all $\ell = 0, \cdots, k$. Then for $\sigma \leq 1/2$,*

$$\left| \log \frac{p * \varphi_{\sigma^2}(x)}{q * \varphi_{\sigma^2}(x)} \right| \leq C\varepsilon \left( \frac{\zeta}{\sigma} \right)^{k+2}, \quad \forall x \in \mathbb{R},$$

*where $\varphi_{\sigma^2}$ is the density function of $\mathcal{N}(0, \sigma^2)$, and $C > 0$ is an absolute constant.*

*Proof.* Write $h = p - q$, then

$$|h * \varphi_{\sigma^2}(x)| = \left| \int_{-\zeta}^{\zeta} h(y) \varphi_{\sigma^2}(x - y) dy \right|$$

$$\overset{(a)}{=} \left| \int_{-\zeta}^{\zeta} h(y) \varphi_{\sigma^2}(x) \sum_{\ell=0}^{\infty} H_\ell \left( \frac{x}{\sigma} \right) \frac{y^\ell}{\ell! \sigma^\ell} dy \right|$$

$$= \left| \sum_{\ell=0}^{\infty} \varphi_{\sigma^2}(x) H_\ell \left( \frac{x}{\sigma} \right) \int_{-\zeta}^{\zeta} h(y) \frac{y^\ell}{\ell! \sigma^\ell} dy \right|$$

$$\overset{(b)}{\leq} 2\varepsilon\zeta \left( \frac{\zeta}{\sigma} \right)^{k+1} \cdot \sum_{\ell=k+1}^{\infty} \frac{\varphi_{\sigma^2}(x)}{\ell!} \left| H_\ell \left( \frac{x}{\sigma} \right) \right|$$

$$\overset{(c)}{\leq} 2\varepsilon\zeta \left( \frac{\zeta}{\sigma} \right)^{k+1} \cdot \sum_{\ell=k+1}^{\infty} \varphi_{\sigma^2}(x) \sum_{m=0}^{\lfloor \ell/2 \rfloor} \frac{(|x|/\sigma)^{\ell-2m}}{(\ell - 2m)! m! 2^m}$$

$$\leq 2\varepsilon\zeta \left( \frac{\zeta}{\sigma} \right)^{k+1} \cdot \sum_{m=0}^{\infty} \frac{\varphi_{\sigma^2}(x)}{m! 2^m} \sum_{\ell=2m}^{\infty} \frac{(|x|/\sigma)^{\ell-2m}}{(\ell - 2m)!}$$

$$\leq 2\varepsilon\zeta \left( \frac{\zeta}{\sigma} \right)^{k+1} \cdot 2\varphi_{\sigma^2}(x) e^{|x|/\sigma}$$

$$= 4e^{1/2}\varepsilon\zeta \left( \frac{\zeta}{\sigma} \right)^{k+1} \cdot \varphi_{\sigma^2}(|x| - \sigma),$$

where $H_\ell(x) = \ell! \cdot \sum_{m=0}^{\lfloor \ell/2 \rfloor} \frac{(-1)^m x^{\ell-2m}}{m!(\ell-2m)!2^m}$ is the Hermite polynomial, (c) uses its analytical form, and (a) uses its exponential generating function:

$$\sum_{\ell=0}^{\infty} H_\ell(x)\frac{t^\ell}{\ell!} = \exp\left(xt - \frac{x^2}{2}\right).$$

As for the step (b), we use the assumed property of $h$ to conclude that

$$\left|\int_{-\zeta}^{\zeta} y^\ell h(y)dy\right| \le 2\varepsilon\zeta^{\ell+1} \cdot \mathbb{1}(\ell > k).$$

On the other hand, to lower bound the denominator $q * \phi_{\sigma^2}(x)$, we have the following observations: as $\sigma \le 1/2$,

$$\int_{-1}^{1} \mathbb{1}(\varphi_{\sigma^2}(x-y) \ge \varphi_{\sigma^2}(|x|-\sigma)/e^2)dy \ge \int_{-1}^{1} \mathbb{1}(\varphi_{\sigma^2}(x-y) \ge \varphi_{\sigma^2}(2\sigma))dy$$

$$\ge \int_{-1}^{1} \mathbb{1}(0 \le y \cdot \text{sign}(x) \le \sigma)dy$$

$$\ge \sigma, \quad \text{if } |x| \le 2\sigma;$$

$$\int_{-1}^{1} \mathbb{1}(\varphi_{\sigma^2}(x-y) \ge \varphi_{\sigma^2}(|x|-\sigma))dy \ge \int_{-1}^{1} \mathbb{1}(\sigma \le y \cdot \text{sign}(x) \le 2\sigma)dy$$

$$\ge \sigma, \quad \text{if } |x| > 2\sigma.$$

Consequently, by Markov's inequality,

$$q * \varphi_{\sigma^2}(x) \ge \frac{1}{4}\int_{-1}^{1} \varphi_{\sigma^2}(x-y)dy$$

$$\ge \frac{\varphi_{\sigma^2}(|x|-\sigma)}{4e^2} \cdot \int_{-1}^{1} \mathbb{1}(\varphi_{\sigma^2}(x-y) \ge \varphi_{\sigma^2}(|x|-\sigma)/e^2)dy$$

$$\ge \frac{\varphi_{\sigma^2}(|x|-\sigma)}{4e^2} \cdot \sigma.$$

A combination of the above inequalities leads to

$$\left|\frac{p * \varphi_{\sigma^2}(x)}{q * \varphi_{\sigma^2}(x)} - 1\right| = \frac{|h * \varphi_{\sigma^2}(x)|}{q * \varphi_{\sigma^2}(x)} \le 16e^{5/2}\varepsilon\left(\frac{\zeta}{\sigma}\right)^{k+2},$$

and therefore the claimed result. $\qquad\square$

Then, we introduce a lemma for constructing two distributions with matched moments.

**Lemma 21.** *Let $\varepsilon > 0$. For any $k = 1, 2, \ldots$, there exists a constant $b > 0$ and a function $h(x)$ supported in $[-\sqrt{\varepsilon}, \sqrt{\varepsilon}]$ such that*

$$\int_0^\infty h(x)dx = \varepsilon, \quad \int h(x)x^i dx = 0, i = 0, 1, \ldots, 2k,$$

*For $i > k$,*

$$\left|\int h(x)x^{2i-1}dx\right| \le 2b\varepsilon^{i+1/2}, \quad \int h(x)x^{2i}dx = 0.$$

*We further have $|h(x)| \le b\sqrt{\varepsilon}$ and $h(x)$ is $b$-Lipschitz continuous. Here the constant $b$ only depends on $k$ and not on $\varepsilon$.*

*Proof.* Note that we only need to prove the lemma for $\varepsilon = 1$, as $h(x) = \sqrt{\varepsilon}h_0(x/\sqrt{\varepsilon})$ only properly scales the moments and preserves Lipschitzness. For $h_0$, consider the following form

$$h_0(x) = \begin{cases} -h_1(-x), & x \in [-1,0] \\ h_1(x), & x \in [0,1] \\ 0, & \text{otherwise} \end{cases},$$

where $h_1(x)$ is a polynomial taking the form

$$h_1(x) = \sum_{i=1}^{k+2} a_i x^i.$$

Let $(a_1, \ldots, a_{k+2})$ be the unique solution to the following linear system:

$$\sum_{i=1}^{k+2} a_i = 0, \quad \sum_{i=1}^{k+2} \frac{a_i}{i+1} = 1, \quad \sum_{i=1}^{k+2} \frac{a_i}{2j+i} = 0, \; j = 1, \ldots, k.$$

Let $b \triangleq \sum_{i=1}^{k+2} i|a_i|$. Then clearly $h_1(0) = h_1(1) = 0$, $|h_1(x)| \leq \sum_{i=1}^{k+2} |a_i| \leq b$, and $|h_1'(x)| \leq \sum_{i=1}^{k+2} i|a_i| \leq b$. It remains to check the odd moments of $h_0$ (all even moments are zero by symmetry). Specifically,

$$\int_0^\infty h_0(x)dx = \int_0^\infty h_1(x)dx = \sum_{i=1}^{k+2} \frac{a_i}{i+1} = 1;$$

$$\int_{-\infty}^\infty h_0(x)x^{2j-1}dx = 2\int_0^\infty h_1(x)x^{2j-1}dx = 2\sum_{i=1}^{k+2} \frac{a_i}{2j+i} = 0, \quad 1 \leq j \leq k;$$

$$\left| \int_{-\infty}^\infty h_0(x)x^{2j-1}dx \right| = 2\left| \int_0^1 h_1(x)x^{2j-1}dx \right| \leq 2b, \quad j > k.$$

This completes the proof. $\qquad\qquad\square$

## F.1 Proof of Lemma 4

For $\sigma \leq c\varepsilon^{1/2}$, we consider two Gaussian distribution $F_1$ as the CDF of $N(0,1)$ and $F_2$ as the CDF of $N(3\varepsilon, 1)$. Then, $g(F_2) - g(F_1) = 3\varepsilon$ and $\mathrm{D_{KL}}(F_1 \| F_2) = O(\varepsilon^2)$. From the data-processing inequality, we have

$$\mathrm{D_{KL}}(F_1 * \mathcal{N}(0,\sigma^2) \| F_2 * \mathcal{N}(0,\sigma^2)) \leq \mathrm{D_{KL}}(F_1 \| F_2) = O(\varepsilon^2).$$

To show the lower bound $\Omega(\varepsilon^2)$, without loss of generality we assume that $F_1^{-1}(0.5) = 0$ and $F_2^{-1}(0.5) \geq \varepsilon$. Proposition 5 and the density lower bound in Assumption 2 show that

$$F_1 * \varphi_{\sigma^2}(0) \geq F_1(0) - \frac{c_2+1}{2}\sigma^2 = \frac{1}{2} - \frac{c_2+1}{2}\sigma^2,$$

$$F_2 * \varphi_{\sigma^2}(0) \leq F_2(0) + \frac{c_2+1}{2}\sigma^2 \leq F_2(\varepsilon) - c_1\varepsilon + \frac{c_2+1}{2}\sigma^2 \leq \frac{1}{2} - c_1\varepsilon + \frac{c_2+1}{2}\sigma^2.$$

Consequently, for $\sigma \leq c\varepsilon^{1/2}$ with a small constant $c > 0$, it holds that

$$F_1 * \varphi_{\sigma^2}(0) - F_2 * \varphi_{\sigma^2}(0) = \Omega(\varepsilon).$$

Therefore, Pinsker's inequality gives

$$\mathrm{D_{KL}}(F_1 * \mathcal{N}(0,\sigma^2) \| F_2 * \mathcal{N}(0,\sigma^2)) \geq \mathrm{D_{TV}}(F_1 * \mathcal{N}(0,\sigma^2) \| F_2 * \mathcal{N}(0,\sigma^2))^2$$
$$\geq |F_2 * \varphi_{\sigma^2}(0) - F_1 * \varphi_{\sigma^2}(0)|^2 = \Omega(\varepsilon^2).$$

For $\sigma \geq \varepsilon^{1/2-\theta}$, let $F_1$ be uniform on $[-1, 1]$. Then, $g(F_1) = 0$. Clearly $F_1$ satisfies Assumption 2. To construct $F_2$, we take the construction of $h(x)$ in Lemma 21 with $k \geq 2\kappa/\theta$ and support on $[-\varepsilon_1, \varepsilon_1]$, with $\varepsilon_1 = b\varepsilon/c_2$. Here $b$ is the Lipschitz constant in Lemma 21, and $c_2$ is the smoothness constant in Assumption 2. The density of $F_2$ is then taken to be

$$p_{F_2}(x) = p_{F_1}(x) + \frac{c_2}{b}h(x).$$

As long as $\varepsilon$ is sufficiently small, we have $|p_{F_2}(x) - 1/2| \leq c_2/b \cdot b\sqrt{\varepsilon_1} \leq 1/4$ everywhere on $x \in [-1, 1]$. In other words, $p_{F_2}(x) \in [1/4, 3/4]$ on its support. Moreover, $p_{F_2}'(x) \leq c_2/b \cdot b = c_2$. This shows that $F_2$ satisfies Assumption 2 as well.

We first show that the median difference between $F_1$ and $F_2$ is at least $\varepsilon$. In fact, by the density upper bound $p_{F_2}(x) \leq 3/4$, we have

$$g(F_2) \geq \frac{4}{3}\left(\frac{1}{2} - F_2(0)\right) = -\frac{4}{3} \cdot \frac{c_2}{b} \int_{-\varepsilon_1}^{0} h(x)dx = \frac{4\varepsilon}{3} > \varepsilon.$$

Next we upper bound the KL divergence between Gaussian convolutions. By choosing $\varepsilon$ sufficiently small, we have $\varepsilon_1 \leq \varepsilon^{1-\theta}$. From Lemma 20 and the property of $h(x)$, we immediately have

$$D_{\mathrm{KL}}(F_1 * \mathcal{N}(0, \sigma^2) \| F_2 * \mathcal{N}(0, \sigma^2))$$

$$\leq \max_{x \in \mathbb{R}} \frac{\log(p_{F_1} * \varphi_{\sigma^2}(x))}{\log(p_{F_2} * \varphi_{\sigma^2}(x))}$$

$$= O\left(\left(\frac{\sqrt{\varepsilon_1}}{\sigma}\right)^{k+2}\right) = O(\varepsilon^{k\theta/2}) = O(\varepsilon^\kappa).$$

## F.2 Proof of Lemma 5

We construct the same pair of distributions $(F_1, F_2)$ as in Lemma 4, and it suffices to prove that when $\sigma \leq c\varepsilon^{1/2}$, we have

$$D_{\mathrm{KL}}(F_2 * \mathcal{N}(0, \sigma^2) \| F_1 * \mathcal{N}(0, \sigma^2)) = \Theta(\varepsilon^{1.5}).$$

For the upper bound, we simply use the data-processing inequality:

$$D_{\mathrm{KL}}(F_2 * \mathcal{N}(0, \sigma^2) \| F_1 * \mathcal{N}(0, \sigma^2)) \leq D_{\mathrm{KL}}(F_2 \| F_1) \leq \chi^2(F_2 \| F_1)$$

$$= \int_{-1}^{1} \frac{(p_{F_1}(x) - p_{F_2}(x))^2}{p_{F_1}(x)} dx$$

$$\leq \left(\frac{c_2}{b}\right)^2 \int_{-1}^{1} \frac{h(x)^2}{1/4} dx$$

$$\leq 4\left(\frac{c_2}{b}\right)^2 \cdot \int_{-\sqrt{\varepsilon_1}}^{\sqrt{\varepsilon_1}} (b\sqrt{\varepsilon_1})^2 dx = O(\varepsilon^{1.5}).$$

For the lower bound, the same proof of Lemma 4 shows that $D_{\mathrm{TV}}(F_1 * \mathcal{N}(0, \sigma^2), F_2 * \mathcal{N}(0, \sigma^2)) = \Omega(\varepsilon)$. A naïve application of Pinsker's inequality only leads to an $\Omega(\varepsilon^2)$ lower bound on the KL divergence. A better lower bound is obtained by noticing that the signed measure $(F_1 - F_2) * \mathcal{N}(0, \sigma^2)$ is effectively supported on $[-\Theta(\sqrt{\varepsilon_1}), \Theta(\sqrt{\varepsilon_1})]$.

To this end, recall from the proof of Lemma 4 that

$$F_1 * \mathcal{N}(0, \sigma^2)(0) - F_2 * \mathcal{N}(0, \sigma^2)(0) = \Omega(\varepsilon).$$

On the other hand, Proposition 5 tells that

$$|F_1 * \mathcal{N}(0, \sigma^2)(-\sqrt{\varepsilon_1}) - F_2 * \mathcal{N}(0, \sigma^2)(-\sqrt{\varepsilon_1})| \leq |F_1(-\sqrt{\varepsilon_1}) - F_2(-\sqrt{\varepsilon_1})| + (c_2 + 1)\sigma^2$$

$$= (c_2 + 1)\sigma^2.$$

Therefore, for $\sigma \leq c\varepsilon^{1/2}$ with a small enough $c > 0$, we have

$$\int_{-\sqrt{\varepsilon_1}}^{0} |p_{F_1} * \mathcal{N}(0, \sigma^2)(x) - p_{F_2} * \mathcal{N}(0, \sigma^2)(x)| dx$$

$$\geq |F_1 * \mathcal{N}(0, \sigma^2)(0) - F_1 * \mathcal{N}(0, \sigma^2)(-\sqrt{\varepsilon_1}) - (F_2 * \mathcal{N}(0, \sigma^2)(0) - F_2 * \mathcal{N}(0, \sigma^2)(-\sqrt{\varepsilon_1}))|$$

$$\geq F_1 * \mathcal{N}(0, \sigma^2)(0) - F_2 * \mathcal{N}(0, \sigma^2)(0) - |F_1 * \mathcal{N}(0, \sigma^2)(-\sqrt{\varepsilon_1}) - F_2 * \mathcal{N}(0, \sigma^2)(-\sqrt{\varepsilon_1})|$$

$$= \Omega(\varepsilon).$$

Let $p(x)$ and $q(x)$ be the shorthands of $p_{F_2} * \mathcal{N}(0, \sigma^2)(x)$ and $p_{F_1} * \mathcal{N}(0, \sigma^2)(x)$, respectively. The KL-divergence can be lower bounded as follows:

$$
\begin{aligned}
\mathrm{D}_{\mathrm{KL}}(F_2 * \mathcal{N}(0, \sigma^2) \| F_1 * \mathcal{N}(0, \sigma^2)) &= \int_{-\infty}^{\infty} p(x) \log \frac{p(x)}{q(x)} dx \\
&= \int_{-\infty}^{\infty} \left( p(x) \log \frac{p(x)}{q(x)} - p(x) + q(x) \right) dx \\
&\overset{(a)}{\geq} \int_{-\sqrt{\varepsilon_1}}^{0} \left( p(x) \log \frac{p(x)}{q(x)} - p(x) + q(x) \right) dx \\
&\overset{(b)}{\geq} \Omega(1) \cdot \int_{-\sqrt{\varepsilon_1}}^{0} (p(x) - q(x))^2 dx \\
&\overset{(c)}{\geq} \Omega(1) \cdot \frac{1}{\sqrt{\varepsilon_1}} \cdot \left( \int_{-\sqrt{\varepsilon_1}}^{0} |p(x) - q(x)| dx \right)^2 \\
&= \Omega(\varepsilon^{1.5}),
\end{aligned}
$$

where

- (a) is due to the non-negativity of $a \log(a/b) - a + b \geq 0$;
- (b) follows from $a \log(a/b) - a + b \asymp (a - b)^2$ whenever $a, b = \Theta(1)$. The latter follows from $|p(x) - p_{F_1}(x)| = O(\sigma) = O(1)$ from Proposition 5, and similarly for $q(x)$;
- (c) makes use of the Cauchy-Schwarz inequality.

This completes the proof.

### F.3 Proof of Theorem 3

From Le Cam's two-point lower bound, it is sufficient to show that the following proposition holds.

**Proposition 10.** *Suppose that $\varepsilon > 0$. Let $\mathcal{F}$ denote the set of distributions satisfying Assumption 2. Consider an online algorithm $\pi$ with a fixed budget $t$ which outputs $\hat{G}$. Given the distribution $F \in \mathcal{F}$ of the underlying arms and the algorithm $\pi$, let $p_{\pi, F}(\{(a_i, y_i)\}_{i=1}^{t})$ denote the distribution of the action-observation pairs up to the $t$-th iteration. Then, for any $\theta \in (0, 1/4)$, there exist $F_1, F_2 \in \mathcal{F}$ with median $g(F_1)$ and $g(F_2)$ such that $|g(F_1) - g(F_2)| \geq \varepsilon$ and*

$$
\mathrm{D}_{\mathrm{KL}}(p_{\pi, F_1} \| p_{\pi, F_2}) \leq O(\varepsilon^{2.5 - \theta} t).
$$

We start with a general log-sum inequality.

**Lemma 22.** *Suppose that $p, q, K$ are probability density functions. Then, we have the following inequality:*

$$
(p * K) \log \frac{p * K}{q * K} \leq \left( p \log \frac{p}{q} \right) * K.
$$

Then, we observe that the pair of distributions $(F_1, F_2)$ constructed in the proof of Lemma 5 satisfies the following property.

**Proposition 11.** *Suppose that $\theta > 0$ is a given constant. Let $\mathcal{F}$ denote the set of distributions satisfying Assumption 2. Then, there exists two distribution $F_1, F_2 \in \mathcal{F}$ with median $g(F_1)$ and $g(F_2)$ such that $g(F_2) = g(F_1) + \varepsilon$ and they satisfy that*

$$
\mathrm{D}_{\mathrm{KL}}(F_1 \| F_2) = O(\varepsilon^{1.5}),
$$

*Denote $\varphi_{\sigma^2}$ as the pdf of $\mathcal{N}(0, \sigma^2)$. For sufficiently small $\varepsilon$ and for $\sigma$ satisfying $\sigma^2 \geq \varepsilon^{1-\theta}$, we further have*

$$
\left| \log \frac{p_{F_1} * \varphi_{\sigma^2}(x)}{p_{F_2} * \varphi_{\sigma^2}(x)} \right| \leq O(\varepsilon^3).
$$

We then continue with the proof of Proposition 10.

*Proof.* Consider two densities defined in Proposition 11 with the parameter $\theta$. Suppose that $x_1, \ldots, x_t$ are i.i.d. samples from either $F_1$ or $F_2$. Then, we note that $a_i \in \mathbb{N}$ for $i \in [t]$ and $y_i \sim \mathcal{N}(x_{a_i}, 1)$ for $i \in [t]$. For simplicity, we write $a^t = (a_1, \ldots, a_t)$ and $y^t = (y_1, \ldots, y_t)$. Hence, we can write the probability distribution of $(a^t, y^t)$ as follows

$$
\begin{aligned}
p_{\pi, F_1}(a^t, y^t) &= \mathbb{E}_{\{x_i\}_{i=1}^t \sim F_1}\left[\prod_{i=1}^t \left(p_\pi(a_i | a^{i-1}, y^{i-1}) \frac{1}{\sqrt{2\pi}} \exp\left(-\frac{1}{2}(y_i - x_{a_i})^2\right)\right)\right] \\
&= \prod_{i=1}^t p_\pi(a_i | a^{i-1}, y^{i-1}) \prod_{j \in \mathbb{N}} \mathbb{E}_{x_j \sim F_1}\left[\prod_{i \leq t, a_i = j} \frac{1}{\sqrt{2\pi}} \exp\left(-\frac{1}{2}(y_i - x_j)^2\right)\right].
\end{aligned}
$$

Let we write $n_j = \sum_{i \leq t} \mathbb{1}(a_i = j)$ and $\bar{y}_j = \frac{1}{n_j} \sum_{i \leq t, a_i = j} y_i$. Then we can write

$$
\mathbb{E}_{x_j \sim F_1}\left[\prod_{i \leq t, a_i = j} \frac{1}{\sqrt{2\pi}} \exp\left(-\frac{1}{2}(y_i - x_j)^2\right)\right] = (p_{F_1} * K_j(\cdot, \bar{y}_j)) \cdot f_j(\{y_i\}_{a_i=j}),
$$

Here we denote

$$
K_j(x, y) = (2\pi)^{-n_j/2} \exp(-n_j(x-y)^2/2)),
$$

and

$$
f_j(\{y_i\}_{a_i=j}) = \exp\left(n_j \bar{y}_j^2/2 - \sum_{i:a_i=j} y_j^2\right).
$$

Therefore, we can write the log-likelihood ratio as

$$
\begin{aligned}
\log \frac{p_{\pi, F_1}(a^t, y^t)}{p_{\pi, F_2}(a^t, y^t)} &= \sum_{j \in \mathbb{N}} \log \frac{\mathbb{E}_{x_j \sim F_1}\left[\prod_{i \leq t, a_i = j} \frac{1}{\sqrt{2\pi}} \exp\left(-\frac{1}{2}(y_i - x_j)^2\right)\right]}{\mathbb{E}_{x_j \sim F_2}\left[\prod_{i \leq t, a_i = j} \frac{1}{\sqrt{2\pi}} \exp\left(-\frac{1}{2}(y_i - x_j)^2\right)\right]} \\
&= \sum_{j \in \mathbb{N}} \log \frac{\mathbb{E}_{x_j \sim F_1}\left[\exp\left(-\frac{n_j}{2}(\bar{y}_j - x_j)\right)\right]}{\mathbb{E}_{x_j \sim F_2}\left[\exp\left(-\frac{n_j}{2}(\bar{y}_j - x_j)\right)\right]}.
\end{aligned}
$$

Then, we can compute that

$$
\begin{aligned}
\mathrm{D}_{\mathrm{KL}}(p_{\pi, F_1} \| p_{\pi, F_2}) &= \int \sum_{a^T} p_{\pi, F_1}(a^T, y^T) \log \frac{p_{\pi, F_1}(a^t, y^t)}{p_{\pi, F_2}(a^t, y^t)} dy^t \\
&= \sum_{j \in \mathbb{N}} \int \sum_{a^T} \prod_{i=1}^t p_\pi(a_i | a^{i-1}, y^{i-1}) \prod_{k \in \mathbb{N}} \mathbb{E}_{x_k \sim F_1}\left[\prod_{i:a_i=k} \frac{1}{\sqrt{2\pi}} \exp\left(-\frac{1}{2}(y_i - x_k)^2\right)\right] \\
&\quad \times \log \frac{\mathbb{E}_{x_j \sim F_1}\left[\exp\left(-\frac{n_j}{2}(\bar{y}_j - x_j)\right)\right]}{\mathbb{E}_{x_j \sim F_2}\left[\exp\left(-\frac{n_j}{2}(\bar{y}_j - x_j)\right)\right]} dy^t.
\end{aligned}
$$

For $n_j \leq \varepsilon^{\theta-1}$, from Proposition 11, we have

$$
\left|\log \frac{F_1 * \mathcal{N}(0, 1/n_i)}{F_2 * \mathcal{N}(0, 1/n_i)}(\bar{y}_j)\right| \leq O(\varepsilon^3).
$$

On the other hand, for $n_j \geq \varepsilon^{\theta-1}$, by utilizing Lemma 22, we note that

$$\int \sum_{a^T} \prod_{i=1}^t p_\pi(a_i|a^{i-1}, y^{i-1}) \prod_{k \in \mathbb{N}} \mathbb{E}_{x_k \sim F_1} \left[ \prod_{i:a_i=k} \frac{1}{\sqrt{2\pi}} \exp\left(-\frac{1}{2}(y_i - x_k)^2\right) \right]$$
$$\times \log \frac{\mathbb{E}_{x_j \sim F_1}\left[\exp\left(-\frac{n_j}{2}(\bar{y}_j - x_j)\right)\right]}{\mathbb{E}_{x_j \sim F_2}\left[\exp\left(-\frac{n_j}{2}(\bar{y}_j - x_j)\right)\right]} dy^t$$
$$= \iint \sum_{a^T} \prod_{i=1}^t p_\pi(a_i|a^{i-1}, y^{i-1}) \prod_{k \neq j} \mathbb{E}_{x_k \sim F_1} \left[ \prod_{i:a_i=k} \frac{1}{\sqrt{2\pi}} \exp\left(-\frac{1}{2}(y_i - x_k)^2\right) \right]$$
$$\times f(\{y_i\}_{a_i=j})[p_{F_1} * K_j(\cdot, \bar{y}_j)](x_j) \log \frac{[p_{F_1} * K_j(\cdot, \bar{y}_j)](x_j)}{[p_{F_2} * K_j(\cdot, \bar{y}_j)](x_j)} dx_j dy^t$$
$$\leq \iint \sum_{a^T} \prod_{i=1}^t p_\pi(a_i|a^{i-1}, y^{i-1}) \prod_{k \neq j} \mathbb{E}_{x_k \sim F_1} \left[ \prod_{i:a_i=k} \frac{1}{\sqrt{2\pi}} \exp\left(-\frac{1}{2}(y_i - x_k)^2\right) \right]$$
$$\times f_j(\{y_i\}_{a_i=j}) p_{F_1}(x_j) \log \frac{p_{F_1}(x_j)}{p_{F_2}(x_j)} K_j(x_j, \bar{y}_j) dx_j dy^t$$
$$= \iint p_{F_1}(x_j) \log \frac{p_{F_1}(x_j)}{p_{F_2}(x_j)} \sum_{a^T} \prod_{i=1}^t p_\pi(a_i|a^{i-1}, y^{i-1})$$
$$\times \int \prod_{k \neq j} \mathbb{E}_{x_k \sim F_1} \left[ \prod_{i:a_i=k} \frac{1}{\sqrt{2\pi}} \exp\left(-\frac{1}{2}(y_i - x_k)^2\right) \right]$$
$$\times \left[ \prod_{i:a_i=j} \frac{1}{\sqrt{2\pi}} \exp\left(-\frac{1}{2}(y_i - x_j)^2\right) \right] dx_j dy^t$$
$$= \int p_{F_1}(x_j) \log \frac{p_{F_1}(x_j)}{p_{F_2}(x_j)} dx_j = O(\varepsilon^{1.5}).$$

Therefore, we have

$$D_{KL}(p_{\pi,F_1}\|p_{\pi,F_2}) \leq \sum_{i \in \mathbb{N}} \left( O(\varepsilon^{1.5})\mathbb{I}(n_i \geq \varepsilon^{\theta-1}) + O(\varepsilon^3)\mathbb{I}(n_i \leq \varepsilon^{\theta-1}) \right).$$

Note that

$$t = \sum_{i \in \mathbb{N}} n_i \geq \varepsilon^{\theta-1} \sum_{i \in \mathbb{N}} \mathbb{I}(n_i \geq \varepsilon^{\theta-1}),$$

and

$$t \geq \sum_{i \in \mathbb{N}} \mathbb{I}(n_i \leq \varepsilon^{\theta-1}).$$

The above inequalities imply that

$$D_{KL}(p_{\pi,F_1}\|p_{\pi,F_2}) \leq O(\varepsilon^{2.5-\theta}t).$$

This completes the proof. $\qquad\square$

### F.4   Proof of Lemma 22

*Proof.* We note that the function $f(x) = x \log x$ is strictly convex. Suppose that $x \in \mathbb{R}$. We note that

$$\int \frac{q(y)K(x-y)}{(K*q)(x)} dy = 1.$$

By the Jensen's inequality, we have

$$\int f\left(\frac{p(y)}{q(y)}\right) \frac{q(y)K(x-y)}{(K*q)(x)} dy \geq f\left(\int \frac{p(y)}{q(y)} \frac{q(y)K(x-y)}{(K*q)(x)} dy\right).$$

This implies that

$$\frac{\left(K * \left(p \log \frac{p}{q}\right)\right)(x)}{(K * q)(x)} \geq \frac{(p * K)(x)}{(q * K)(x)} \log \frac{(p * K)(x)}{(q * K)(x)}.$$

This completes the proof.

$\square$

# G   Proof of Lower Bounds for trimmed mean estimation

## G.1   Proof of Lemma 6

Firstly, for $\sigma \leq C\varepsilon^{1/2}$, we consider two Gaussian distribution $F_1$ as the CDF of $N(0,1)$ and $F_2$ as the CDF of $N(3\varepsilon, 1)$. Then, $g(F_2) - g(F_1) = 3\varepsilon$ and $D_{\mathrm{KL}}(F_1 \| F_2) = \frac{9\varepsilon^2}{2}$. From the data-processing inequality, we have

$$D_{\mathrm{KL}}(F_1 * \mathcal{N}(0, \sigma^2) \| F_2 * \mathcal{N}(0, \sigma^2)) \leq D_{\mathrm{KL}}(F_1 \| F_2) = O(\varepsilon^2).$$

To show the lower bound $\Omega(\varepsilon^2)$, without loss of generality we assume that $\int_{F_1^{-1}(\alpha)}^{F_1^{-1}(1-\alpha)} x \, dF_1(x) = 0$ and $\int_{F_2^{-1}(\alpha)}^{F_2^{-1}(1-\alpha)} x \, dF_2(x) \leq -\varepsilon$. Lemma 9 and the density lower bound in Assumption 4 show that

$$\int_{(F_1 * \varphi_{\sigma^2})^{-1}(\alpha)}^{(F_1 * \varphi_{\sigma^2})^{-1}(1-\alpha)} x \, dF_1 * \varphi_{\sigma^2}(x) \geq \int_{F_1^{-1}(\alpha)}^{F_1^{-1}(1-\alpha)} x \, dF_1(x) - \tilde{C}\sigma^2 = -\tilde{C}\sigma^2,$$

$$\int_{(F_2 * \varphi_{\sigma^2})^{-1}(\alpha)}^{(F_2 * \varphi_{\sigma^2})^{-1}(1-\alpha)} x \, dF_2 * \varphi_{\sigma^2}(x) \leq \int_{F_2^{-1}(\alpha)}^{F_1^{-1}(1-\alpha)} x \, dF_2(x) + \tilde{C}\sigma^2 \leq -\varepsilon - \tilde{C}\sigma^2,$$

Here $\tilde{C} > 0$ is a constant. Consequently, for $\sigma \leq c\varepsilon^{1/2}$ with a small constant $c > 0$, it holds that

$$\int_{(F_1 * \varphi_{\sigma^2})^{-1}(\alpha)}^{(F_1 * \varphi_{\sigma^2})^{-1}(1-\alpha)} x \, dF_1 * \varphi_{\sigma^2}(x) - \int_{(F_2 * \varphi_{\sigma^2})^{-1}(\alpha)}^{(F_2 * \varphi_{\sigma^2})^{-1}(1-\alpha)} x \, dF_2 * \varphi_{\sigma^2}(x) = \Omega(\varepsilon).$$

From the algorithm for trimmed mean estimation, we can distinguish $F_1 * \varphi_{\sigma^2}$ and $F_2 * \varphi_{\sigma^2}$ using $O(\varepsilon^{-2})$ samples. This implies that $D_{\mathrm{KL}}(F_1 * \mathcal{N}(0, \sigma^2) \| F_2 * \mathcal{N}(0, \sigma^2)) \geq \Omega(\varepsilon^2)$.

For $\sigma \geq \varepsilon^{1-\theta}$, let $F_1$ be uniform on $[1, 2]$. Without the loss of generality, we may assume that $\sigma^2 \geq 4$, $c_1 \leq 0.5$, $c_3 \geq 2$, $c_4 \geq 2$, $c_5 \leq 1$. Then, $F_1$ satisfies Assumption 4. By taking $k \geq \frac{2\kappa}{\theta}$, we can construct $h(x)$ satisfying the conditions in Lemma 21 with $\varepsilon_1 = 4b'\varepsilon$, where $b' = \max\{\frac{b}{c_2}, 4\}$. Consider the distribution $F_2$ with pdf $p_{F_2}(x) = p_{F_1}(x) + (b')^{-1}h(x - 1 - \alpha)$. By choosing $\varepsilon$ sufficiently small such that $\sqrt{\varepsilon_1} \leq \min\{\alpha, 1 - 2\alpha\}$, the density function $p_{F_2}$ is supported in $[1, 2]$ and $F_2^{-1}(1 - \alpha) = 2 - \alpha$.

As $|h(x)| \leq b\sqrt{\varepsilon_1}$, for $\varepsilon_1 \leq \frac{1}{4c_2^2}$, we have $(b')^{-1}h(x) \leq \frac{c_2}{b}h(x) \leq c_2\sqrt{\varepsilon} \leq \frac{1}{2}$. This implies that $p_{F_2}(x) \in [1/2, 3/2]$ for $x \in [1, 2]$. Therefore, $p_{F_2}$ is a density function. Because $p'_{F_1}(x) = 0$ for $x \in [1, 2]$, $p_{F_2}(x)$ is $c_2$-Lipschitz continuous in $[1, 2]$.

Note that $F_2(1 + \alpha) = F_1(1 + \alpha) + (b')^{-1}\int_{-\infty}^{0} h(x) dx = \alpha - 4\varepsilon$. Hence, it follows that

$$4\varepsilon = F_2(F_2^{-1}(\alpha)) - F_2(1 + \alpha) \leq 3/2(F_2^{-1}(\alpha) - 1 - \alpha).$$

This implies that $F_2^{-1}(\alpha) \geq 1 + \alpha + \frac{2}{3} \cdot 4\varepsilon \geq 1 + \alpha + 2\varepsilon$. Therefore,

$$\int_{F_2^{-1}(\alpha)}^{F_2^{-1}(1-\alpha)} x \, dF_2(x) \leq \int_{1+\alpha+\varepsilon}^{2-\alpha} x \, dF_2(x) = 2 - 2\alpha - 2\varepsilon - \int_{\varepsilon}^{\sqrt{\varepsilon_1}} (b')^{-1}h(x) dx$$

$$\leq 2 - 2\alpha - 2\varepsilon + 2\varepsilon\sqrt{(b')^{-1}} \leq 2 - 2\alpha - \varepsilon.$$

Here we utilize that $b' \geq 4$. Note that $\int_{F_1^{-1}(\alpha)}^{F_1^{-1}(1-\alpha)} x \, dF_1(x) = 2 - 2\alpha$. This implies that

$$g(F_2) - g(F_1) \leq -\varepsilon.$$

By choosing $\varepsilon$ sufficiently small, we have $\varepsilon_1 \leq \varepsilon^{1-\theta/2}$. From Lemma 20 and the property of $h(x)$, we immediately have

$$D_{\mathrm{KL}}(F_1 * \mathcal{N}(0, \sigma^2) \| F_2 * \mathcal{N}(0, \sigma^2)) \leq \max_{x \in \mathbb{R}} \frac{\log(p_{F_1} * \varphi_{\sigma^2}(x))}{\log(p_{F_2} * \varphi_{\sigma^2}(x))}$$

$$\leq O(1) \cdot \frac{b\sqrt{\varepsilon_1}\sqrt{\varepsilon_1}^{2k+2}}{\sigma^{2k+2}} = O(\varepsilon^{(2k+1)\theta/2}) = O(\varepsilon^{\kappa}).$$

This completes the proof.

## G.2  Proof of Lemma 7

We construct the same pair of $F_1$ and $F_2$ as in Lemma 6. It is sufficient to prove that when $\sigma \leq c\varepsilon^{1/2}$,

$$D_{\mathrm{KL}}(F_2 * \mathcal{N}(0, \sigma^2) \| F_1 * \mathcal{N}(0, \sigma^2)) = \Theta(\varepsilon^{1.5}).$$

For the upper bound, we simply use the data-processing inequality:

$$D_{\mathrm{KL}}(F_2 * \mathcal{N}(0, \sigma^2) \| F_1 * \mathcal{N}(0, \sigma^2)) \leq D_{\mathrm{KL}}(F_2 \| F_1) \leq \chi^2(F_1 \| F_2)$$

$$= \int_1^2 \frac{(p_{F_1}(x) - p_{F_2}(x))^2}{p_{F_1}(x)} dx$$

$$\leq (b')^2 \int_{-\sqrt{\varepsilon_1}}^{\sqrt{\varepsilon_1}} h(x)^2 dx$$

$$\leq (b')^2 \cdot \int_{-\sqrt{\varepsilon_1}}^{\sqrt{\varepsilon_1}} (b\sqrt{\varepsilon_1})^2 dx = O(\varepsilon^{1.5}).$$

For the lower bound, we note that $F_2(1 + \alpha) = \alpha - 4\varepsilon = F_1(1 + \alpha) - 4\varepsilon$. Similar to the proof of Lemma 5, we have

$$F_1 * \mathcal{N}(0, \sigma^2)(1 + \alpha) - F_2 * \mathcal{N}(0, \sigma^2)(1 + \alpha) = \Omega(\varepsilon).$$

Analogously, we can derive the same lower bound

$$D_{\mathrm{KL}}(F_2 * \mathcal{N}(0, \sigma^2) \| F_1 * \mathcal{N}(0, \sigma^2)) \geq \Omega(\varepsilon^{3/2}).$$

## G.3  Proof of Theorem 4

By applying the Le Cam's two point lower bound, it is sufficient to show that the following proposition holds.

**Proposition 12.** *Suppose that $\varepsilon > 0$. Denote $\mathcal{F}$ as the set the set of distributions satisfying Assumption 4. Consider an online algorithm $\pi$ with a fixed budget $t$ which outputs $\hat{G}$. Given the distribution with CDF $F \in \mathcal{F}$ of the underlying arms and the algorithm $\pi$, let $p_{\pi,F}(\{(a_i, y_i)\}_{i=1}^t)$ denote the distribution of the action-observation pairs up to the $t$-th iteration. Then, for any $\theta \in (0, 1/4)$, there exist $F_1, F_2 \in \mathcal{F}$ with trimmed means $g(F_1)$ and $g(F_2)$ such that $|g(F_1) - g(F_2)| \geq \varepsilon$ and*

$$D_{\mathrm{KL}}(p_{\pi,F_1} \| p_{\pi,F_2}) \leq O(\varepsilon^{2.5-2\theta} t).$$

Similar to the proof of Proposition 10, we start with the following proposition.

**Proposition 13.** *Suppose that $\theta > 0$ is a given constant. Let $\mathcal{F}$ denote the set of distributions satisfying Assumption 4. Then, there exists two distribution $F_1, F_2 \in \mathcal{F}$ with trimmed mean $g(F_1)$ and $g(F_2)$ such that $g(F_2) \leq g(F_1) - \varepsilon$ and they satisfy that*

$$D_{\mathrm{KL}}(F_1 \| F_2) = O(\varepsilon^{1.5-\theta}),$$

*Denote $\varphi_{\sigma^2}$ as the pdf of $\mathcal{N}(0, \sigma^2)$. For sufficiently small $\varepsilon$ and for $\sigma$ satisfying $\sigma^2 \geq \varepsilon^{1-\theta}$, we further have*

$$\left| \log \frac{p_{F_1} * \varphi_{\sigma^2}(x)}{p_{F_2} * \varphi_{\sigma^2}(x)} \right| \leq O(\varepsilon^3).$$

Then, we present the proof of Proposition 12.

*Proof.* Consider $F_1$ and $F_2$ as distributions constructed in Proposition 13. Based on Proposition 13 and Lemma 22, analogously, we have

$$\mathrm{D}_{\mathrm{KL}}(p_{\pi,F_1} \| p_{\pi,F_2}) \leq \sum_{i \in \mathbb{N}} \left( O(\varepsilon^{1.5-\theta}) \mathbb{I}(n_i \geq \varepsilon^{\theta-1}) + O(\varepsilon^3) \mathbb{I}(n_i \leq \varepsilon^{\theta-1}) \right).$$

Note that

$$t = \sum_{i \in \mathbb{N}} n_i \geq \varepsilon^{\theta-1} \sum_{i \in \mathbb{N}} \mathbb{I}(n_i \geq \varepsilon^{\theta-1}), \quad t \geq \sum_{i \in \mathbb{N}} \mathbb{I}(n_i \leq \varepsilon^{\theta-1}).$$

This implies that

$$\mathrm{D}_{\mathrm{KL}}(p_{\pi,F_1} \| p_{\pi,F_2}) \leq O(\varepsilon^{2.5-2\theta} t).$$

This completes the proof. $\qquad\square$

## G.4 Proof of Proposition 13

Consider two distributions constructed in Lemma 6. It is sufficient to show the bound on KL divergence and the pointwise bound. Firstly, we note that $F_1$ is uniform on $[1, 2]$ and the density of $F_2$ only differs from the density of $F_1$ in $[1 + \alpha - \sqrt{\varepsilon_1}, 1 + \alpha + \sqrt{\varepsilon_1}]$. The difference is upper bounded by $b\sqrt{\varepsilon_1}$. We note that for sufficiently small $\varepsilon$, we have $\varepsilon_1 \leq \varepsilon^{1-\theta/2}$. Therefore, we have

$$\begin{aligned}
\mathrm{D}_{\mathrm{KL}}(F_1 \| F_2) &= -\int_{1+\alpha-\sqrt{\varepsilon_1}}^{1+\alpha+\sqrt{\varepsilon_1}} p_{F_1} \log \frac{p_{F_2}}{p_{F_1}} dx \\
&= -\int_{1+\alpha-\sqrt{\varepsilon_1}}^{1+\alpha+\sqrt{\varepsilon_1}} p_{F_1} \left( \left( \frac{p_{F_2}}{p_{F_1}} - 1 \right) - \frac{1}{2} \left( \frac{p_{F_2}}{p_{F_1}} - 1 \right)^2 + O(\varepsilon_1^{3/2}) \right) dx \\
&= \frac{1}{2} \int_{1+\alpha-\sqrt{\varepsilon_1}}^{1+\alpha+\sqrt{\varepsilon_1}} \left( \frac{p_{F_2}}{p_{F_1}} - 1 \right)^2 dx + O(\varepsilon^2) \\
&= O(\varepsilon_1^{\frac{3}{2}}) = O(\varepsilon^{\frac{3}{2}(1-\theta/2)}) = O(\varepsilon^{1.5-\theta}).
\end{aligned}$$

We note that $F_1$ and $F_2$ have $2k$ matched moments, where $k > 6/\theta$. Analogous to the result in Lemma 20, for all $x \in \mathbb{R}$,

$$\left| \log \frac{p_{F_1} * \varphi_{\sigma^2}(x)}{p_{F_2} * \varphi_{\sigma^2}(x)} \right| \leq C \frac{b\sqrt{\varepsilon_1}\sqrt{\varepsilon_1}^{-2k+2}}{\sigma^{2k+2}} \leq C \cdot \frac{b\varepsilon^{(k+1)(1-\theta/2)}}{\varepsilon^{(k+1)(1-\theta)}} = O(\varepsilon^{(k+1)\theta/2}) \leq O(\varepsilon^3).$$

Here $C > 0$ is an absolute constant. This completes the proof.