# OpenReview forum: "Beyond the Best:  Distribution Functional Estimation in Infinite-Armed Bandits"
_NeurIPS.cc/2022/Conference — NeurIPS 2022 Accept_

### Official Review · Reviewer_9e61 · 2022-07-08

**Rating:** 5
**Confidence:** 4
**Soundness:** 4 excellent
**Presentation:** 3 good
**Contribution:** 3 good

**Summary:**

The authors study the infinite-armed bandit problem. Prior work focused on best arm identification in this setting; the authors instead consider estimating a functional of the underlying distribution F. They examine indicator-based functionals such as the mean, median, maximum, and trimmed mean. They study both the offline and online settings. They give a meta-algorithm for arbitrary indicator-based functionals with a general sample complexity and also provide lower bound results for each setting.

**Questions:**

Could the authors comment on the optimality of the algorithms in all of the settings in more detail?

Could the authors elaborate on the motivation more? As of now, the applications are unclear to me.

Post rebuttal: the authors have addressed my questions on optimality, theoretical comparison between the offline and online settings, and the motivation.

**Limitations:**

Yes.

**Strengths And Weaknesses:**

Strengths:

It is nice that they have a unifying algorithm and sample complexity result for all settings.

They provide lower bounds for all of the settings.

The problem formulation is mathematically elegant and it is interesting to consider indicator-based functionals.

Weaknesses:

The motivation for estimating a general functional seems weak to me. The authors mention single-cell RNA sequencing, but do not explain to the reader in detail how estimating the median or trimmed mean arise in these practical settings. I think it is particularly important to motivate estimating the median and trimmed mean because these seems like the most novel settings of this paper. In what real-world problems would we apply these algorithms? I also think the offline setting should be motivated here.

I also think the paper would benefit from experimentally validating the results. How much of a gap is there between the online and offline algorithms in practice?

The algorithmic novelty is limited. While it is nice that the algorithm unify all settings, it uses standard ideas from the literature, applied to the indicator-based functional. It basically adaptively identifies which drawn arms belong to the set defined by the indicator-based functional using confidence bounds and order statistics.

It would be nice to give some concrete examples where Theorem 2 gives a much better sample complexity than is possible in the online setting.

I think that there could be a clearer presentation relating the upper bounds and the lower bounds. The authors argue that the bounds are tight, but for the online setting upper bounds contain a distance^2 term that does not appear in the lower bounds. So, it is difficult to evaluate the gap.

The algorithm seems to require knowing the quantities from the assumptions in section 3.1. These do not seem like they would typically be known in practice. It would be useful and important to give guidance to practitioners on how to set these quantities in practice.

---

> ### Author Response · Authors · 2022-08-02
> **Response to Reviewer 9e61**
>
> We thank the reviewer for their comments and feedback. We have striven to incorporate their feedback, and hope that the below clarifications address your concerns.
>
> __{Motivation of functional estimation:__ we thank the reviewer for pointing this out. We have added further motivation in the introduction, including a practical distributed learning application (see general response for additional discussion). We have also added additional motivation regarding the offline setting, as while adaptivity is incredibly powerful it is not always feasible, due to the length of time required to obtain samples as in biological experiments or the communication overhead in distributed learning.
>
> __Algorithmic novelty:__ see general response.
>
> __Theorem 2:__ Theorem 2 details our algorithm's performance in the online setting. We present the corollaries of Theorem 2 in appendix D.3, and provide worst-case bounds for Theorem 2 in Table 1, showing that median and trimmed mean estimation is improved by a factor of $\epsilon^{-.5}$ by our online algorithm as opposed to the offline baseline, for example.
>
> __Tightness of bounds (distance squared term):__ the lower bounds we present in this work are worst-case, not instance dependent. That is to say, from a lower bound perspective, we show that there exist 2 hard distributions that any adaptive algorithm must draw many samples from in order to provide $(\epsilon,.1)$-PAC estimation. We provide an instance-dependent upper bound for our algorithm in Theorem 2, and show in Appendix D.3 that for any distribution satisfying the functional-specific assumptions in Section 3.1, the sample complexity of the online algorithm will be no worse than the minimax lower bound (up to terms suppressed by $\tilde{\Theta}$).
>
> __Assumptions in Section 3.1:__ In order to exploit the Bayesian nature of the problem, these assumptions are necessary. Concretely, in the maximum estimation setting, an upper bound of $\beta$ is required in order to provide any theoretical guarantees. Adapting to $\beta$ (e.g. if the provided upper bound is much larger than the true $\beta$) is known to be difficult, and requires significant problem specific work [Simple regret for infinitely many armed bandits, 2015]. Thus, in this first work tackling general functional estimation in the bandit setting, it does not seem to be helpful to discuss this adaptivity, and upper bounds of these quantities should be used. Note that our algorithm doesn't require exact knowledge of these distributional parameters, and simply requires bounds on these quantities (e.g. Lipschitz continuity around quantiles).
>
> __Optimality of the algorithms:__ Our algorithm sample complexities are detailed in Table 1, showing that our algorithms are minimax optimal up to lower order terms suppressed by $\tilde{\Theta}$. Our upper bounds for the offline setting are detailed in Theorem 1, and in the online setting are described in Theorem 2. Corollary 2.2 gives the lower bound for mean estimation, Corollary 2.3 gives the lower bound for maximum estimation, Corollary 2.4 gives the lower bound for offline median estimation, and Theorem 3 gives the lower bound for online median estimation. Corollary 3.1 and Theorem 4 provide the lower bound for trimmed mean estimation.

---

### Official Review · Reviewer_YnSc · 2022-07-10

**Rating:** 7
**Confidence:** 3
**Soundness:** 3 good
**Presentation:** 3 good
**Contribution:** 4 excellent

**Summary:**

The paper studies distribution functional estimation (estimating mean, median, maximum, trimmed mean) in the infinite-armed bandit problem.

For the offline setting, the authors design the appropriate number of points $n$ and the number of samples per point $m$ to minimize the sample complexity $m*n$.

For the online setting, they propose an elimination-based algorithm that can adaptively screen out those redundant points, which are not related to the functional estimation. Thus, the online algorithm can reduce the sample complexity (in expectation). However, the online algorithm shares the same worst-case sample complexity as the offline algorithm.

They also provide matching lower bounds for the sample complexity under online/offline settings. For the mean and maximum estimation, the proof uses Wasserstein distances to upper bound the KL divergence.  For the median estimation, the Wasserstein distance-based method is too loose, so they propose a proof and show a ``thresholding phenomenon’’, KL divergence does not change smoothly with the noise level, which may be of independent interest.


**Questions:**

Note that the sample complexity depends on the target accuracy $\epsilon$ also the confidence level $\delta$. Why not consider the dependence on $\delta$ in Table 1? Is that because all the terms have the same growth rate $O(log(1/\delta))$?

Since the objective in this paper is no longer minimizing regret or identifying the best arm as in the classical MAB setting, could the author further list the related work in median/quantile estimation? For example, the result for mean estimation is a simple extension of concentration/Chebyshev inequality. For the online setting, there’s no need to discard points, since every point should contribute to the mean.

**Strengths And Weaknesses:**

Originality: It’s a novel work. The paper focuses on the sample complexity and reveals the differences between various functions and offline/online settings. My only concern is that the work seems to be more related to statistics, but the paper doesn't review papers in that area. I think a discussion of the connection to papers in statistics can help the position of the paper a lot.

Quality: It’s technically sound.

Clarity: The paper is written clearly and well organized.

Significance: The paper considers the sample complexity for different functionals and online/offline settings. It provides matching upper bound and lower bound for every scenario. The result (which is summarized in Table 1) is novel, complete, and solid.

The paper shows interesting insights which can shed some light to future work. First, the work reveals the difficulty for various functions. For example, estimating the median is harder than the mean, but not harder than the trimmed mean. Second, the work reveals the difference between offline and online settings. For mean estimation, the online setting has no advantage. But online algorithm has improved over offline for other functionals.

---

> ### Author Response · Authors · 2022-08-02
> **Response to Reviewer YnSc**
>
> We thank the reviewer for their careful reading and positive feedback. We have worked to clarify related statistical works, and have incorporated additional discussion regarding the error probability $\delta$.
>
> __Related work in statistics:__ as the reviewer points out, there are indeed relevant statistical works. However, these works have only considered the offline setting, as the observation model motivating the online setting is novel and specific to the multi-armed bandit setting. We now include these references and additional background in the main text, see the general response for further discussion.
>
> __Dependence on $\delta$:__ indeed, the results in Table 1 depend on $\delta$. Since the focus of this work was on characterizing the dependence on $\epsilon$, $\delta$ was taken to be a constant for this table (see lower bounds, $\delta$ taken to be $.1$). Currently, our approaches have differing dependencies on $\delta$ due to the different assumptions and analysis techniques. Detailed dependence on $\delta$ can be found in Proposition 1-4 for the offline algorithm and Theorem 2 for the online algorithm. For instance, mean estimation uses $n\propto 1/\delta$ from Prop 1, as the Chebyshev inequality is used. By further assuming that the underlying distribution is sub-Gaussian, we can improve this dependence to $n\propto \log(1/\delta)$. Maximum estimation in the offline setting uses $n,m\propto \log(1/\delta)$, requiring a total sample complexity scaling with $\log^2(1/\delta)$. Observe from Theorem 2 that our online algorithm will only increase the expected sample complexity by at most a $\log(1/\delta)$ factor as compared to the offline method (however, the online sample complexity can be no worse than the offline one). Considering the simultaneous limiting behavior as $\epsilon,\delta \to 0$ is an interesting direction of future work.

---

### Official Review · Reviewer_246j · 2022-07-11

**Rating:** 5
**Confidence:** 3
**Soundness:** 3 good
**Presentation:** 3 good
**Contribution:** 3 good

**Summary:**

The paper considered distribution functional estimation problems in the infinite armed bandit problem.
First, the authors formulated the distribution functional estimation problems (especially mean, median, maximum, trimmed mean estimations).
The authors presented meta-algorithms for such distribution functional estimation problems for both online and offline settings together with the sample complexity guarantees.
Next, lower bounds on the sample complexities for both offline and online settings are shown.
For the lower bound for the median estimation problem, the authors remarked on "thresholding phenomena" in order to prove tighter lower bounds.


**Questions:**

Please see the Limitations section.

**Limitations:**


My main concern is that the research associated with this paper is not well surveyed.
As the main objective is different from the papers cited in the related work, I don't think the paper deals with the problems that could only be cast as bandit problems.
The problem considered in this paper (estimating the distribution functional estimation in online/offline setting) and the problem considered in the related work (minimizing regret or
simple regret) have a largely different nature.

It looks like there are many papers in statistics that deals with the problem of estimating the statistical functionals. for example,

Wasserman, Larry. All of statistics: a concise course in statistical inference. Vol. 26. New York: Springer, 2004.

Hall, Peter, and Soumendra N. Lahiri. "Estimation of distributions, moments and quantiles in deconvolution problems." The Annals of Statistics 36.5 (2008): 2110-2134.

(Note these areas are not my expertise.)

While there may be a novelty, I believe that appropriate comparisons with papers dealing with similar issues need to be made.






**Strengths And Weaknesses:**

- Strengths

A relatively thorough upper/lower bound characterization of the distribution functional estimation problem has been made. (for both offline and online settings)


- Weaknesses

The online algorithm is based on the doubling trick so it may be unpractical. (although theoretically sound)
No numerical experiments have been made.
Positioning of the work (see Limitations section).

---

> ### Author Response · Authors · 2022-08-02
> **Response to Reviewer 246j**
>
> We thank the reviewer for their careful reading. We have worked to better place our work in the existing literature by adding a paragraph comparing our work with existing statistical literature. As we describe, the offline setting has been previously considered (non-adaptive), but the adaptivity inherent to the online infinite-armed bandit setting has not been studied.
>
> __Doubling trick:__ the choice of per-round arm budgets is a minor component in our algorithm, but we note that the "doubling trick" normally refers to regret-based algorithms, where a fixed-budget algorithm which attains $O(\log T)$ regret for a fixed horizon $T$ is converted to an algorithm with anytime performance guarantees by restarting it from scratch in epochs of progressively doubling length. The reviewer is correct that such methods often perform poorly in practice due to the discarding of all prior information in each new epoch. Our algorithm does not use this approach as our objective is identification, not regret minimization. While our number of arm pulls does indeed increase geometrically across rounds, this form of doubling has been shown to empirically yield good performance ([Almost Optimal Exploration in Multi-Armed Bandits, 2013], [Distributed Exploration in Multi-Armed Bandits 2013]). This scaling of arm pulls is just for convenience, trading off between rounds of adaptivity and number of arm pulls required, and any increasing sequence can be used (e.g. one pull per round).
>
> __Relation with existing statistical literature:__ we thank the reviewer for highlighting this important point. We have sought to better place our work in the context of the existing statistical estimation literature, and have expanded upon this point in the general response.

---

> > ### Author Response · Authors · 2022-08-08
> > **Response to Reviewer 246j**
> >
> > Dear Reviewer 246j,
> >
> > We thank you for your valuable time spent reviewing our work, and we really hope to have a further discussion with you to see if our response resolved your concerns. Within the rebuttal period, we have made the following improvements based on your suggestions, which have been integrated into the latest version:
> >
> > - We add further motivation in the introduction.
> > - We better place our work in the context of the existing statistical estimation literature.
> >
> > We would appreciate it if you could kindly share your thoughts on the key points in our response, and keep the discussion rolling in case you have further comments. Thank you!
> >
> > Best wishes,
> >
> > Authors

---

> > > ### Comment · Reviewer_246j · 2022-08-09
> > > **Re: Response to Reviewer 246j**
> > >
> > > Many thanks for your detailed comments and modification of your draft.
> > > I understand your use of the Doubling trick and its relevance to the related research. Although I still believe the work shouldn't be reviewed solely from the bandit's perspective, I acknowledge the contribution. I raised the score to 5.
> > >
> > > Best,
> > >
> > > Reviewer 246j

---

### Official Review · Reviewer_Go45 · 2022-07-25

**Rating:** 6
**Confidence:** 3
**Soundness:** 3 good
**Presentation:** 2 fair
**Contribution:** 2 fair

**Summary:**

The paper studies the problem of functional estimation under the infinite-armed bandit setting (Berry et al., 1997). Specifically, the authors considered online and offline algorithms for estimating multiple functionals, such as mean, median, maximum, and trimmed mean. The main contribution is determining the sample complexity bounds for these estimation tasks.

**Questions:**

Please refer to the strengths and weaknesses section for suggestions or points to address.

A question here is whether the authors think some of these ideas apply to the classical bandit settings, and why.

**Limitations:**

I think the settings and results are reasonable. But I didn't find a specific section or paragraph discussing the limitations of the current work or future directions. The authors may consider adding such a section since there's still space.

**Strengths And Weaknesses:**

One strength is that the sample complexity bounds are tight, up to logarithmic factors in the error parameters.

Another strength is that the authors have unified formulations and algorithms for both settings, showing that the online and offline sample complexities can differ for some functions. One can find more details in Table 1.

A weakness, in my opinion, is that the paper does not provide much motivation for learning functionals (other than the mean) under the infinite-armed bandit setting. I found one sentence on page 1 saying that "in many scenarios, including single-cell RNA-sequencing (Zhang et al., 2020)", but it doesn't seem that a related experiment appeared in the paper.

Another significant weakness is the writing, especially the clarity. I suggest the authors introduce the critical definitions and formulations before explaining the results. In the paper, the authors seem to assume that the readers are already familiar with the related literature.

One more weakness is the novelty. The offline algorithm is an averaging algorithm with standard concentration attributes. And the online algorithm essentially refines the rough ordering of elements by constructing finer and finer confidence intervals on the promising ones. I don't think these ideas are novel, or maybe they are relatively standard.

---

> ### Author Response · Authors · 2022-08-02
> **Response to Reviewer G045**
>
> We thank the reviewer for their careful reading of the paper. We have added text to the manuscript to describe the novelty and scope of the setting in greater detail, and have clarified the motivation of decoupling the observation model and the statistical objective.
>
> __Motivation:__ we have added additional background and motivation to the introduction, as detailed in the general response.
>
> __Clarity:__ thanks to the reviewer's comments we have worked to further clarify the exposition by expanding on the definitions and formulations prior to the results.
>
> __Novelty:__ while the offline algorithm does not allow for many algorithmic degrees of freedom, the analysis required is novel, especially for the accompanying lower bound. Further, while the online setting may appear similar to existing bandit works, the key algorithmic aspect of leveraging information across arms to exploit the underlying Bayesian structure is novel. See the general response for further points.
>
> __Applicability to classical bandit setting:__ We think that these ideas are applicable only for the infinite-armed scenario, and not the classical setting. This highlights the novelty in our setting; due to the common underlying arm reward distribution, in order to estimate the median of the arm-reward distribution, we do not need to identify the median arm. This ability to average over arms is unique to the Bayesian nature of the infinite-armed bandit problem.
>
> __Limitations:__ Interesting directions of future work include extending our analysis to non-indicator-based functionals, such as the BH threshold. We have added this to the conclusion in the revised manuscript.

---

> > ### Author Response · Authors · 2022-08-08
> > **Response to Reviewer G045**
> >
> > Dear Reviewer G045,
> >
> > We thank you for your valuable time spent reviewing our work, and we really hope to have a further discussion with you to see if our response resolved your concerns. Within the rebuttal period, we have made the following improvements based on your suggestions, which have been integrated into the latest version:
> >
> > - We add further motivation in the introduction.
> > - We elaborate on the algorithmic novelty and theoretical novelty in the lower bound analysis.
> > - We better place our work in the context of the existing statistical estimation literature.
> > - We discuss the future directions of the extension of our analysis.
> >
> > We would appreciate it if you could kindly share your thoughts on the key points in our response, and keep the discussion rolling in case you have further comments. Thank you!
> >
> > Best wishes,
> >
> > Author

---

### Author Response · Authors · 2022-08-02
**Overall Response**

__Motivation:__
In this work, we showed how the infinite-armed bandit setting can be utilized to study more general functionals beyond the maximum, decoupling the observation model and the objective.

The infinite-armed bandit observation model has received much attention due to its applicability in large-scale settings [Bandit problems with infinitely many arms, 1997]. Objectives beyond the maximum are commonly of interest; e.g. conditional value at risk, quantiles, and beyond [PAC identification of a bandit arm relative to a reward quantile, 2017] [Quantile-Regret Minimisation in Infinitely Many-Armed Bandits, 2018].

In this work, we show that this observation model can be extended beyond just the case of maximum estimation, and construct a general algorithm with fundamental theoretical guarantees that can be applied to many settings. One such natural setting is in large-scale distributed learning. Here, a server / platform wants to estimate how much test-users like their newly released product. Users return a noisy realization of their affinity for the product, and the platform can decide to pay the user further to spend more time with the product, to provide additional testing.

For many natural objectives which are robust to a small fraction of adversarial users, e.g. trimmed mean, median, or quantile estimation, we see that our algorithm will enable estimation of the desired quantity to high accuracy while minimizing the total cost (number of samples taken). We include the following sentences discussing the motivation of learning general functionals in the revised manuscript.

In [Adaptive Monte Carlo Multiple Testing via Multi-Armed Bandits, 2019], estimating the Benjamini Hochberg (BH) threshold arises in the field of multiple hypothesis testing with applications in statistical inference. The estimation of the median/quantile is similar to estimation of the BH threshold, as both depend on the order statistics of the underlying distribution.  Estimating the median or trimmed mean has important applications in robust statistics, for instance, maintaining the fidelity of the estimator in the presence of outliers. We also emphasize that the median estimation in our setting is different from the usual median estimation setting. This is because each sample is noisy but in the statistical literature we have clean samples whenever we draw one sample from the underlying distribution.

__Relation with existing literature:__
Following the reviewers' helpful comments, we have added the following discussion to the manuscript to better position our work.

From a statistical perspective, the sample complexity in the offline setting is closely related to deconvolution distribution estimation [Deconvolution of a distribution function. 1997] [All of statistics: a concise course in statistical inference. 2004] [Estimation of distributions, moments and quantiles in deconvolution problems. 2008] [On deconvolution with repeated measurements. 2008] [On deconvolution of distribution functions, 2011]. We cite these references and elaborate on the below comparison in the revision. Nevertheless, these previous works mainly focus on the expected L2 difference between the underlying distribution function and its estimation. This simplified setting does not allow for consideration of the trade-off inherent in our setting between the number of points and the (variable) number of observations per point. Additionally, these past works did not calculate the specific sample complexity for functionals like median and quantile. Since the noise is treated as fixed and uniform, there has been no study of the online setting where adaptive resampling enables dramatic sample complexity improvements. In particular, the challenge is that we have noisy observations, which makes the lower bound even in offline cases a significant challenge that has not been dealt with in the past, let alone the online case.

---

> ### Author Response · Authors · 2022-08-02
> **Overall Response (continued)**
>
> __Novelty:__
> While our proposed algorithms share surface-level similarity with existing algorithms for the classical finite-arm bandit problem, our algorithm has significant algorithmic and analytical differences from classical bandits. From an algorithmic perspective, we need to carefully prioritize arms based on their confidence intervals in order to determine which arms may still impact the quantity of interest, and then leverage the Bayesian nature of the problem by averaging those arms determined to have means within the range of interest.
> From an analytical perspective, we cannot directly show that our online algorithm achieves $(\epsilon,\delta)$-PAC estimation, but rather need to first show that our online algorithm provides $(\epsilon/2,\delta/2)$-PAC estimation of the offline algorithm, which we then show achieves $(\epsilon/2,\delta/2)$-PAC estimation of the desired functional. This first step shares similarities with Frequentist analyses, and the second with Bayesian analyses and functional estimation ones in statistics, but both require nontrivial modifications for this specific setting. Observe that in standard Frequentist settings, one cannot average the estimated arm-mean across different arms (as arm means are arbitrary), and for maximum estimation in the Bayesian setting averaging is not beneficial. Studying functionals beyond the maximum in the Bayesian setting necessitates this more sophisticated algorithmic structure which we propose.
>
> In addition to the novel techniques required for the algorithm design and analysis, the lower bounds provided in this work are novel and revealing. Even the lower bounds we provide in the offline case differ significantly from the usual problems considered in the statistics literature. Further, we utilize the Wasserstein distance to prove lower bounds for both offline and online algorithms, and discover a thresholding phenomenon of the Wasserstein distance with respect to the noise variance arising in median estimation, which may be of independent interest.

---

### Meta-Review · Area_Chair_GEt9 · 2022-08-20

**Recommendation:** Accept
**Confidence:** Less certain

**Metareview:**

This paper studies offline and online statistical estimation in the infinite-armed bandit setting and gives a set of almost tight upper and lower bounds on the sample complexity.  Initially, some reviewers raised concerns about the motivation of general functional estimation and the comparison with existing statistical estimation literature. The authors have made efforts on addressing these issues; their arguments seem reasonable.

**Award:**

No

---

### Decision · Program_Chairs · 2022-09-14

Accept